# **BRIGHT:** A globally distributed multimodal building damage assessment dataset with very-high-resolution for all-weather disaster response

Hongruixuan Chen<sup>1,2</sup>, Jian Song<sup>1,2</sup>, Olivier Dietrich<sup>3</sup>, Clifford Broni-Bediako<sup>2</sup>, Weihao Xuan<sup>1,2</sup>, Junjue Wang<sup>1</sup>, Xinlei Shao<sup>1</sup>, Yimin Wei<sup>1</sup>, Junshi Xia<sup>2</sup>, Cuiling Lan<sup>4</sup>, Konrad Schindler<sup>3</sup>, and Naoto Yokoya<sup>1,2</sup>

<sup>1</sup>Graduate School of Frontier Sciences, The University of Tokyo, Chiba, Japan

<sup>2</sup>RIKEN Center for Advanced Intelligence Project (AIP), RIKEN, Tokyo, Japan

<sup>3</sup>Department of Photogrammetry and Remote Sensing, ETH Zürich, Zürich, Switzerland

<sup>4</sup>Microsoft Research Asia, Beijing, China

Correspondence: Naoto Yokoya (yokoya@k.u-tokyo.ac.jp)

**Abstract.** Disaster events occur around the world and cause significant damage to human life and property. Earth observation (EO) data enables rapid and comprehensive building damage assessment, an essential capability crucial in the aftermath of a disaster to reduce human casualties and inform disaster relief efforts. Recent research focuses on developing artificial intelligence (AI) models to accurately map unseen disaster events, mostly using optical EO data. These solutions based on optical

- data are limited to clear skies and daylight hours, preventing a prompt response to disasters. Integrating multimodal EO data, particularly combining optical and synthetic aperture radar (SAR) imagery, makes it possible to provide all-weather, day-and-night disaster responses. Despite this potential, the lack of suitable benchmark datasets has constrained the development of robust multimodal AI models. In this paper, we present a <u>Building damage assessment dataset using veRy-hIGH-resoluTion</u> optical and SAR imagery (BRIGHT) to support AI-based all-weather disaster response. To the best of our knowledge, BRIGHT
- is the first open-access, globally distributed, event-diverse multimodal dataset specifically curated to support AI-based disaster response. It covers five types of natural disasters and two types of human-made disasters across 14 regions worldwide, focusing on developing countries where external assistance is most needed. The dataset's optical and SAR images with spatial resolutions between 0.3 and 1 meters provide detailed representations of individual buildings, making it ideal for precise damage assessment. We train seven advanced AI models on BRIGHT to validate transferability and robustness. Beyond that, it
- also serves as a challenging benchmark for a variety of tasks in real-world disaster scenarios, including unsupervised domain adaptation, semi-supervised learning, unsupervised multimodal change detection, and unsupervised multimodal image matching. The experimental results serve as baselines to inspire future research and model development. The dataset (Chen et al., 2025), along with the code and pretrained models, is available at https://github.com/ChenHongruixuan/BRIGHT and will be updated as and when a new disaster data is available. BRIGHT also serves as the official dataset for the 2025 IEEE GRSS Data
- Fusion Contest Track II. We hope that this effort will promote the development of AI-driven methods in support of people in disaster-affected areas.

## 1 Introduction

- A disaster is defined as a severe disruption in the functioning of a community or society due to the interaction between a hazard event and the conditions of exposure, vulnerability and capacity resulting in human, material, economic or environmental losses and impacts (Ge et al., 2020). According to the United Nations Office for Disaster Risk Reduction (UNDRR), between 1998 and 2017, natural disasters such as earthquakes, storms, and floods affected approximately 4.4 billion people and caused 1.3 million deaths. These disasters have also resulted in economic losses of US\$2,647 billion in disaster-affected countries (UNDRR, 2018a). The threat of disasters is likely to increase due to global urbanization (Kreibich et al., 2022; Bastos Moroz and Thieken, 2024). Rapid and comprehensive damage assessment is crucial in the aftermath of a disaster to make informed and effective rescue decisions that minimize losses and impacts. Building damage assessment aims to provide information, including the area and amount of damage, the rate of collapsed buildings, and the type of damage each building has incurred.
  - This information is critical in the early stages of a disaster, as the distribution of damaged buildings is closely related to lifesaving efforts in an emergency response (Xie et al., 2016; Adriano et al., 2021). Conducting field surveys after a disaster can be difficult and dangerous, especially when transportation and communication systems are disrupted, making efficient on-site
- 35 assessments challenging. Earth observation (EO) provides a safe and efficient way to obtain information on building damage in disaster areas due to its wide field of view and contactless operation.

The EO technologies commonly used for assessing building damage after disasters are optical and synthetic aperture radar (SAR). Optical imagery is a primary source for building damage assessment because of its intuitive and easy-to-interpret nature. For example, moderate-resolution optical data from the Landsat series and Sentinel-2 have been used to assess building

damage (Yusuf et al., 2001; Fan et al., 2019; Sandhini Putri et al., 2022). Landsat and Sentinel-2 data are limited in spatial resolution and only provide broad approximations of affected areas, which lack precision for specific buildings, crucial for timely rescue. The new generation of very high-resolution (VHR) optical sensors, such as IKONOS and WorldView, provides EO data with spatial resolutions of a meter or less, enabling finer assessments at the level of individual buildings (Freire et al., 2014). These data have been used successfully in building damage assessment (Yamazaki and Matsuoka, 2007; Tong et al., 2012; Freire et al., 2014).

While accurate building damage maps can be obtained by visual interpretation of optical images by human experts, this process is time-consuming and labor-intensive for large-scale rapid assessments. In addition, it requires trained professionals. Therefore, recent studies have focused on developing automated methods for rapid building damage mapping (Tong et al., 2012; Xie et al., 2016; Gupta et al., 2019; Zheng et al., 2021). Among these, machine learning (ML) and deep learning (DL)

- techniques have significantly improved efficiency and accuracy in building damage assessment. Earlier work focused on a single disaster event with labels annotated for a specific disaster area to train a model. This model is then used to generate building damage maps for the same event (Xie et al., 2016; Xia et al., 2023). However, since training data were limited to a few building types, damage patterns, and background land cover distributions, the resulting models mostly lack generalizability and struggle to produce accurate building damage maps for new disaster events, which limits their practical use. Recent large-scale
- benchmark datasets, for example, the xBD dataset (Gupta et al., 2019) containing different types of disaster scenarios and

(a) Pre-event optical imagery

(b) Post-event optical imagery

(c) Post-event SAR imagery

**Figure 1.** An example of the wildfire occurring in Maui, Hawaii, USA, August 2023. (a) Pre-event optical imagery (© Maxar). (b) Post-event optical image (© Maxar) with land-cover features obscured by wildfire smoke. (c) Post-event SAR imagery (© Capella Space) unaffected by smoke, showing the disaster area.

damages, have made it possible to adopt DL models to quickly and accurately map building damages after a newly occurred, previously unseen disaster (Zheng et al., 2021; Chen et al., 2022a; Shen et al., 2022; Kaur et al., 2023; Guo et al., 2024; Wang et al., 2024; Chen et al., 2024). For example, Zheng et al. (2021) trained DL models on the xBD dataset and applied them to map the damage to buildings in two unseen human-made disaster events. These studies have demonstrated the effectiveness of DL models for building damage mapping.

DL models for building damage mapping.

The optical EO technology uses a passive sensing technique, which requires solar illumination and cloud-free weather conditions. This severely limits the application of optical images in an emergency tool for all-weather disaster response (Adriano et al., 2021). In contrast, SAR sensors employ active illumination with longer microwaves and can acquire images in adverse weather conditions, offering great potential for all-weather disaster response. Most disaster events, especially wildfires, floods,

and storms, are often accompanied by less-than-ideal imaging conditions. For example, Figure 1 shows EO imagery captured for a wildfire event that occurred in August 2023 in Hawaii, USA. The post-event optical image shown in Figure 1-(b) does not provide clear surface information due to the effects of the wildfire smoke. However, the SAR image illustrated in Figure 1-(c) is not affected by smoke and clearly shows the buildings damaged by the wildfire.

Due to the advantages of SAR imagery, various SAR-based methods have been proposed for building damage assessment. 70 These methods utilize intensity (Matsuoka and Yamazaki, 2005, 2010; Matsuoka et al., 2010), coherence (Yonezawa and Takeuchi, 2001; Arciniegas et al., 2007; Watanabe et al., 2016; Liu and Yamazaki, 2017), and polarization features (Yamaguchi, 2012; Chen and Sato, 2013; Watanabe et al., 2016; Karimzadeh and Mastuoka, 2017) to assess building damage at a block unit level, depending on the acquisition mode. Several studies have attempted to extend the block-level approach and have explored new approaches at the building instance level using higher spatial resolution sensors such as COSMO-SkyMed and TerraSAR-

75 X (Liu et al., 2013; Brett and Guida, 2013; Chini et al., 2015; Ge et al., 2019). DL-based methods have also been explored