# Peer review of "BRIGHT: A globally distributed multimodal building damage assessment dataset with very-high-resolution for all-weather disaster response"

_Earth System Science Data, 2025_

## Author Comment (AC1)

**Response to Comments of Referee #1**

Thank you for the instructive and constructive comments for our paper. Those comments are very helpful for and serve as significant guidance for our research. We have studied the comments carefully and revised our manuscript accordingly. The changes in our manuscript are highlighted in **red**. The point-to-point responses to your questions/comments are listed as follows.

**Comments to the Author:**

This paper introduces BRIGHT, a novel and timely benchmark dataset for building damage assessment using multimodal high-resolution optical and SAR imagery. Covering 14 globally distributed disaster events, BRIGHT provides pixel-level damage annotations for over 384,000 buildings. The dataset is designed to facilitate AI-based disaster response research, particularly in challenging all-weather conditions. The authors also benchmark a suite of machine learning and deep learning models on multiple tasks. The authors provided detailed documents and descriptions, making the data, related source code, and pretrained weights of models easy to understand and use.

In summary, this is quite interesting and solid work. I'd like to recommend the acceptance of this work since it represents an important contribution to Earth observation and disaster response communities. Yet before acceptance, several clarifications and refinements are suggested.

**Response:** We really appreciate your spot-on summary of our manuscript and such a positive endorsement of our work. Our responses to your valuable comments and suggestions are itemized below.

**Q1:** The manuscript would benefit from deeper exploration of what the models learn from multimodal fusion. Specifically, what roles do optical images play in multimodal building damage assessment? Is it beyond just building footprint localization? On the other words, are the features extracted from optical imagery actively compared with SAR representations? Some discussion (e.g., based on CAMs in Fig. 7) is provided but can be more explicitly elaborated.

**R1:** Thank you so much for this very insightful comment. To investigate the role of optical imagery in multimodal building damage assessment, we conducted additional experiments as suggested. Specifically, we evaluated UNet and DeepLabV3+ under two input settings: optical + SAR and SAR only. We chose these two models because they are a single-branch architecture, making it straightforward to adjust the number of input channels by modifying the first convolutional layer. In contrast, the other five methods adopt Siamese networks, where structural changes for different input modalities would require extensive reconfiguration. For UNet and DeepLabV3+, the modification introduces negligible

changes in the parameter count. To isolate the contribution of optical imagery beyond building footprint localization, we provided all models with perfect building masks as post-processing steps prior to evaluation.

The results, presented in **Table 7** of the revised manuscript, demonstrate that optical imagery contributes significantly to distinguishing different damage levels. When provided with optical + SAR inputs, both models show notable improvements in the IoU scores for the "Damaged" and "Destroyed" classes compared to SAR-only inputs. For example, UNet's IoU for "Damaged" improved from 35.83% to 44.83%. DeepLabV3+ also benefits from optical imagery, with IoU for "Damaged" changing from 39.63% (SAR only) to 40.45% (optical + SAR), and for "Destroyed" increasing substantially from 59.54% to 64.94%. These findings indicate that optical imagery provides critical complementary information that supports damage classification, rather than merely improving building localization.

Accompanying Table 7, we have added a new **Section 5.3** to the revised manuscript to provide a more detailed discussion of these findings. We show the revised part below for your convenience.

**Table 7.** Comparison of UNet and DeepLabV3+ performance using only SAR and optical-SAR inputs for damage classification. Here, accurate building masks are provided as the post-processing step to all models to isolate the effect of building localization on the damage classification task.

| Method     | Modality    | $F_1^{clf}$ (%) | Final mIoU (%) | IoU per class (%) |        |         |           |  |
|------------|-------------|-----------------|----------------|-------------------|--------|---------|-----------|--|
|            | liteanity   | -1 (/0)         |                | Background        | Intact | Damaged | Destroyed |  |
| UNet       | SAR         | 68.71           | 69.84          | 100.0             | 88.19  | 35.83   | 55.35     |  |
|            | Optical-SAR | 73.59           | 72.41          | 100.0             | 89.38  | 44.83   | 55.42     |  |
| DeepLabV3+ | SAR         | 72.12           | 72.19          | 100.0             | 89.59  | 39.63   | 59.54     |  |
|            | Optical-SAR | 73.90           | 73.93          | 100.0             | 90.32  | 40.45   | 64.94     |  |

**450 5.3 The role of optical pre-event data in multimodal building damage mapping**

In the last section, CAM visualizations revealed that DL models also exhibit responses to disaster-specific patterns in pre-event optical imagery. This observation suggests that optical data may play a more complex role in multimodal building damage mapping than simply supporting building localization. In other words, in a multimodal bi-temporal setup, does pre-event optical imagery act solely as a localization aid, or does it provide additional semantic cues that networks can exploit for more accurate damage classification?

**455 accurate damage classification?**

To explore this, we conducted controlled experiments using UNet and DeepLabV3+. Both networks were trained under two configurations: (1) using post-event SAR imagery only, and (2) using multimodal pre- and post-event inputs (optical-SAR). To isolate the contribution of pre-event optical data beyond building localization, we provided perfect building masks for postprocessing in both settings. This design ensures that any observed differences in performance are attributable to the additional information from pre-event optical imagery, rather than differences in network architecture or localization accuracy. The results, summarized in Table 7, show that incorporating pre-event optical imagery leads to notable improvements in distinguishing building damage levels. For UNet, the IoU for the "Damaged" class increased from 35.83% (SAR only) to 44.83% (Optical-SAR), and for the "Destroyed" class from 55.35% to 55.42%. DeepLabV3+ exhibited significant gains also, with IoU improvements from 39.63% to 40.45% for "Damaged" category, and from 59.54% to 64.94% for "Destroyed" category. These

**465** results suggest that pre-event optical imagery contributes beyond mere building localization, enriching the feature space for more effective semantic comparison for different building damage levels across modalities.

**Q2:** The manuscript makes extensive evaluations of supervised and unsupervised change detection models, but the conceptual and methodological relationship between building damage assessment and generic change detection remains unclear, which is largely implied rather than discussed. An explicit and clearer explanation would be great for readers who lack of related background.

**R2:** Thank you for this insightful comment. We agree that clarifying the conceptual and methodological relationship between building damage assessment (BDA) and generic change detection (CD) will help readers unfamiliar with the field.

Specifically, a common view is to treat BDA as a special case of "one-to-many" semantic change detection tasks [1]-[4], where the goal is to assess not just whether a change has occurred but also to characterize the type and severity of the change (*i.e.*, levels of damage). In this sense, BDA extends beyond binary change detection by requiring finer-grained semantic interpretation of pre- and post-event imagery. Many existing methods for BDA are thus derived from or adapted versions of generic change detection models. Furthermore, in some unified change detection frameworks [3]-[5], BDA is explicitly included as one of the downstream tasks, highlighting their methodological overlap.

It is important to note that this discussion focuses on the formulation of BDA tasks that take bitemporal inputs (i.e., both pre- and post-disaster images). Alternative approaches that rely solely on post-disaster imagery exist but are outside the scope of our evaluation and discussion.

We have added the above description in Section 4.1 of the revised manuscript to clarify this problem. We show the revised part below for your convenience.

- [1] Z. Zheng, Y. Zhong, J. Wang, A. Ma, and L. Zhang, "Building damage assessment for rapid disaster response with a deep object-based semantic change detection framework: From natural disasters to man-made disasters," *Remote Sensing of Environment*, vol. 265, p. 112636, 2021.
- [2] W. Lu, L. Wei and M. Nguyen, "Bitemporal Attention Transformer for Building Change Detection and Building Damage Assessment," *IEEE Journal of Selected Topics in Applied Earth Observations* and Remote Sensing, vol. 17, pp. 4917-4935, 2024.
- [3] H. Chen, J. Song, C. Han, J. Xia and N. Yokoya, "ChangeMamba: Remote Sensing Change Detection with Spatiotemporal State Space Model," *IEEE Transactions on Geoscience and Remote Sensing*, vol. 62, pp. 1-20, 2024.
- [4] Z. Zheng, Y. Zhong, J. Zhao, A. Ma, and L. Zhang, "Unifying remote sensing change detection via deep probabilistic change models: From principles, models to applications," *ISPRS Journal of Photogrammetry Remote Sensing*, vol. 215, pp. 239–255, 2024.
- [5] H. Guo, X. Su, C. Wu, B. Du and L. Zhang, "SAAN: Similarity-Aware Attention Flow Network for Change Detection with VHR Remote Sensing Images," *IEEE Transactions on Image Processing*, vol. 33, pp. 2599-2613, 2024.

are employed in the experiments to compare their results.

It is worth noting that in this work, we focus on the formulation of building damage assessment as a bi-temporal task, where both pre- and post-event images are used as inputs. This formulation aligns closely with generic change detection tasks, which aim to identify changes between two time points. Conceptually, building damage assessment can be viewed as a specialized "one-to-many" semantic change detection problem (Zheng et al., 2021, 2024; Lu et al., 2024), where the objective is not only to detect whether a change has occurred but also to categorize the type and severity of changes (damages) to buildings. Many existing methods are thus derived from or adapted versions of generic change detection frameworks (Chen et al., 2024; Zheng et al., 2024; Guo et al., 2024).

4.2 Benchmark suites

**Q3:** Since UMCD methods underperform, consider including a random guessing baseline for reference. This would contextualize the difficulty of BRIGHT and help readers understand the performance floor under UMCD setup.

**R3:** Thank you for your insightful suggestion. We have added the results of a random guessing baseline to **Table 12** for reference. As shown below, the different methods achieve improvements over random guessing; however, the gains are not very significant. This highlights the challenging nature of applying UMCD methods to the BRIGHT dataset. We show the revised part below for your convenience.

**Table 12.** Results of representative unsupervised multimodal change detection methods. KC is the acronym of kappa coefficient. The highest values are highlighted in **bold**, and the second-highest results are highlighted in underline. The accuracies on the UMCD benchmark dataset are the accuracies on the four datasets presented in Figure G1, obtained from their literature. Details of methods and benchmark datasets are presented in Appendix G. The random guessing baseline is included to indicate the performance floor under the UMCD setup. The "-" symbol indicates that the corresponding method did not report results on that dataset in their original publications.

| Method                       | UMCD benchmark datasets |                         |                        |              | Bright |      |             |  |
|------------------------------|-------------------------|-------------------------|------------------------|--------------|--------|------|-------------|--|
|                              | OA                      | F1                      | КС                     | OA           | F1     | IoU  | KC          |  |
| Random guessing              | 50.0                    | 8.4 / 6.0 / 11.0 / 11.4 | 0.0                    | 50.00        | 7.83   | 4.08 | 0.00        |  |
| IRG-McS (Sun et al., 2021)   | 98.3 / - / 97.1 / 97.2  | 80.4 / - / 75.4 / 73.7  | 79.4 / - / 73.9 / 75.1 | 90.03        | 12.65  | 6.75 | 7.65        |  |
| SR-GCAE (Chen et al., 2022b) | 98.6 / 98.5 / - / -     | 82.9 / 77.6 / - / -     | 82.1 / 76.9 / - / -    | 77.83        | 14.35  | 7.73 | 5.64        |  |
| FD-MCD (Chen et al., 2023)   | 98.2 / 97.8 / - / 96.7  | 81.4 / 72.2 / - / 73.2  | 82.3 / 71.1 / - / 71.4 | 80.96        | 15.84  | 8.60 | 7.94 |  |
| AOSG (Han et al., 2024)      | - / - / - / 96.4        | -/-/-/77.7              | -/-/75.9               | 77.93        | 10.75  | 5.68 | 3.98        |  |
| AGSCC (Sun et al., 2024a)    | 98.3 / - / 95.9 / 97.6  | 78.2 / - / 68.0 / 77.9  | 77.3 / - / 65.8 / 76.6 | 88.49 | 14.82  | 8.00 | 9.54        |  |
| AEKAN (Liu et al., 2025)     | 98.7 / - / - / 98.3     | 83.8 / - / - / 84.7     | 83.1/ - / - / 83.9     | 81.60        | 13.09  | 7.00 | 3.56        |  |

**Q4:** While Table 1 offers a comprehensive comparison of datasets, several datasets seem relevant and should be included to enhance its completeness, like CRASAR-U-DROIDs [arXiv:2407.17673] and Noto-Earthquake building damage dataset [10.5194/essd-2024-363].

**R4:** Thank you for your valuable suggestion. We have reviewed the CRASAR-U-DROIDs [arXiv:2407.17673] and the Noto-Earthquake Building Damage Dataset [10.5194/essd-2024-363] and

**have updated Table 1 to include them for a more comprehensive comparison.**

Table 1. Comparison of BRIGHT with the existing building damage assessment datasets. The OA indicates whether the dataset is open access (OA) or not, and GSD is an acronym for ground sampling distance (GSD). Note that since some datasets integrate other datasets, we summarize only the largest one to avoid duplication here. For example, the BDD dataset (Adriano et al., 2021) includes the Tohoku-Earthquake-2011 dataset (Bai et al., 2018) and Palu-Tsunami-2018 dataset (Adriano et al., 2019). No. of building Dataset OA Modality GSD (m/pixel) No. of events Disaster type Granularity ABCD (Fujita et al., 2017) Optical EO 0.4 Tsunami N/A Image-level (Nguyen et al., 2017) N/A 4 3 natural disasters N/A Image-level  $\checkmark$ Images on social media (Cheng et al., 2021) Optical EO 1,802 Image-level  $\checkmark$ N/A 1 Hurricane (Xue et al., 2024) Street-view image N/A 1 Hurricane 2.468 Image-level FloodNet (Rahnemoonfar et al., 2021) Optical EO N/A 1 Flood 6,675 Pixel-level RescueNet (Rahnemoonfar et al., 2023) Optical EO N/A 1 Hurricane 10,903 Pixel-level Ida-BD (Kaur et al., 2023) Optical EO 18,083 Pixel-level 0.5 1 Hurricane **CRASAR-U-DROIDs** 4 natural disasters Optical EO 0.02-0.12 10 21,716 Pixel-level 1 (Manzini et al., 2024) 1 man-made disaster Noto-BDA-MV (Vescovo et al., 2025) **Optical EO** N/A 1 Earthquake 140.208 Pixel-level xBD (Gupta et al., 2019) Optical EO < 0.8 15 6 natural disasters >700,000 Pixel-level QQB (Sun et al., 2024b) Optical and SAR EO

We show the corresponding revised part below for your convenience.

**Q8:** 10: Add a note in the caption to clarify that each dot corresponds to performance on a single test event under cross-event transfer.

**R8:** Thank you for your thoughtful comment. We believe you were referring to **Figure 10**. We have added a note in the caption to clarify that each dot represents the performance on a single test event under cross-event transfer.

We show the corresponding revised part below for your convenience.

---

## Author Comment (AC3)

Thank you for the instructive and constructive comments for our paper. Those comments are very helpful for and serve as significant guidance for our research. We have studied the comments carefully and revised our manuscript accordingly. The changes in our manuscript are highlighted in **red**. The point-to-point responses to your questions/comments are listed as follows.

**Comments to the Author:**

This manuscript introduces the BRIGHT dataset, which is the first open building damage assessment dataset with global coverage, multi-hazard scenarios, multimodal imagery (Optical and SAR), and sub-meter resolution. The paper systematically describes data collection, annotation, and quality control methods, and validates the dataset with multiple deep learning models, including cross-disaster transfer (zero-shot and one-shot), semi-supervised, and unsupervised approaches. The dataset demonstrates clear novelty and practical value, and it is of significant importance for advancing research and applications in disaster emergency response, remote sensing, and artificial intelligence. Generally, the paper is well structured, logically clear, with detailed results and strong value in terms of data sharing. Although the manuscript is rich in content, there are still details that require improvement, and I recommend appropriate revisions.

**Response:** We really appreciate your spot-on summary of our manuscript and such a positive endorsement of our work. Our responses to your valuable comments and suggestions are itemized below.

**Q1:** The explanation of annotation consistency and reliability remains insufficient. Although the authors state that the data annotations were obtained from multiple institutions such as Copernicus EMS, UNOSAT, and FEMA and then refined manually, there may be inconsistencies in how different institutions define "damaged" and "destroyed". This could affect the consistency of annotations across disaster scenarios. It is therefore necessary to further elaborate on the process of unifying annotations, provide more detail on the manual refinement procedures.

**R1:** We sincerely thank the reviewer for raising this crucial point. Ensuring annotation consistency across different data sources and disaster types is paramount for the reliability of BRIGHT as a benchmark dataset, and we appreciate the opportunity to elaborate on our rigorous unification and refinement process.

Our approach was a multi-stage process designed specifically to address the potential inconsistencies the reviewer has identified:

First, we recognized that the source agencies, while conceptually aligned, use slightly different grading scales. To address this, we **established a single, standardized three-tier classification scheme for all events** in BRIGHT: Intact (1), Damaged (2), and Destroyed (3), with clear definitions provided in Table 3 of our manuscript. This scheme served as the universal target for all incoming annotations.

Secondly, the reviewer correctly notes that the exact terminology and number of damage tiers can differ between agencies. However, **their underlying definitions for EO-based damage assessment are conceptually consistent**. All agencies grade damage based on visually verifiable structural failure. This conceptual alignment provided a solid foundation for our initial, rule-based mapping. The "Destroyed" category was the most consistent. Labels such as "Destroyed", "Collapsed", or "Completely Damaged" from all sources were directly mapped to our Destroyed (3) class. For partial damage, we aggregated multiple intermediate tiers. Labels like "Severe Damage", "Major Damage", "Highly Damaged", or "Moderately Damaged" were all mapped to our single Damaged (2) class. This conservative aggregation ensures that our "Damaged" category represents significant, visually verifiable structural harm.

Recognizing that subtle inconsistencies could persist even after the rule-based mapping, the most critical stage of our process was a comprehensive manual review and refinement. This final, expert-led stage served as the ultimate guarantor of consistency, ensuring that every annotation conforms to our unified standard. This procedure, conducted using tools like Google Earth Pro, involved:

➢ Correction of Inconsistencies: Our experts meticulously compared pre- and post-disaster VHR optical imagery for each annotation point to identify and correct discrepancies between the source label and the visual evidence.

➢ Harmonization of Ambiguous Labels: We paid special attention to **ambiguous source labels**, such as "Possibly Damaged". In these cases, if clear structural damage was not evident upon visual inspection, we adopted a conservative approach and re-classified the building as "Intact" to ensure a high confidence "Damaged" class.

➢ Disaggregation of Area-Based Labels: Crucially, we **identified and re-processed all area-based damage annotations** (i.e., where an entire block or neighborhood was assigned a single damage category). Our team manually disaggregated these coarse labels, assigning a precise, building-wise (point-level) damage label to each individual structure within the area. This step was vital for ensuring instance-level consistency and granularity across the entire dataset.

Through this systematic process of standardization, mapping, and exhaustive expert-led refinement, we have made every effort to harmonize the annotations and ensure that the final labels in BRIGHT are as consistent and reliable as possible. We have now added these details to the manuscript to make our process more transparent.

Here, we show the revised part below for your convenience.

The labels in Bright consist of two components: building polygons and post-disaster building damage attributes. Expert annotators manually labeled the building polygons, then all labels underwent independent visual inspections of EO experts to ensure accuracy. Damage annotations were obtained from Copernicus Emergency Management Service[6], the United Nations Satellite Centre (UNOSAT) Emergency Mapping Products[7], and the Federal Emergency Management Agency (FEMA)[8]. These annotations were derived through visual interpretation of high-resolution optical imagery captured before and after the disasters by EO experts, supplemented by partial field visits. To harmonize these diverse annotations and ensure consistency across all 14 disaster events, we implemented a rigorous, multi-stage process. First, we established a single, standardized three-tier classification scheme, including Intact (with pixel value 1), Damaged (with pixel value 2), and Destroyed (with pixel value 3), with clear definitions provided in Table 3, drawing on the frameworks of FEMA's Damage Assessment Operations Manual, EMS-98, the BDD dataset (Adriano et al., 2021), and the xBD dataset (Gupta et al., 2019). While the source agencies' terminology can differ (e.g., "Severe Damage" vs. "Major Damage"), their underlying definitions for EO-based assessment are conceptually consistent. We leveraged this alignment for an initial rule-based mapping, where various intermediate damage tiers were conservatively aggregated into our single "Damaged" category. Second, our team of EO experts conducted a comprehensive manual verification and refinement of every annotation using multi-temporal VHR imagery on platforms like Google Earth Pro. This final stage served as the ultimate guarantor of consistency. We paid special attention to ambiguous source labels, such as "Possibly Damaged". Adopting a conservative approach, these were re-classified as "Intact" if clear structural damage was not evident, thereby ensuring a high-confidence "Damaged" class. We also manually disaggregated all area-based annotations (i.e., where an entire block was assigned a single category). We re-processed these to assign a precise, building-wise damage label to each individual structure, ensuring instance-level consistency and granularity across the entire dataset.

[6]https://emergency.copernicus.eu
[7]https://unosat.org/products
[8]https://www.fema.gov

**Q2:** The treatment of class imbalance is not sufficient. Figure 5(d) shows that intact buildings account for over 80%, while destroyed buildings account for less than 7%. This severe imbalance directly affects the accuracy of recognizing destroyed classes. Although the authors employed the Lovasz loss function to partially alleviate the issue, this is still not enough to solve the problem. Is this imbalance one of the reasons for the relatively low performance of the subsequent experimental results?

**R2:** We thank the reviewer for this insightful question. The reviewer has astutely identified one of the most significant and inherent challenges in the task of automated building damage assessment.

First, we would like to clarify that this severe class imbalance is not a unique artifact of our dataset but rather **a common characteristic of real-world post-disaster data**. Disasters, even when severe, typically damage or destroy a minority of the buildings in an affected area. For instance, the widely used **xBD dataset** exhibits a similar long-tail distribution, with the "intact" class constituting the vast majority of labeled buildings **(approximately 75%)**. The imbalance in BRIGHT, as shown in Figure 5(d), therefore realistically reflects the sparse nature of catastrophic damage.

Second, we agree with the reviewer that this imbalance is indeed one of the reasons for the lower performance observed in our benchmark results. Unlike general land-cover mapping tasks, where classes tend to be more balanced, the **rarity and variability of damage signatures** make it especially difficult for models to learn robust and generalizable representations from a limited number of disaster

events.

Then, we wish to clarify the primary scope of our work. As a dataset and benchmark paper, **our central contribution is to capture and present these real-world challenges** in multimodal building damage mapping, including the severe class imbalance, to **provide a realistic and challenging testbed for the community**. Our objective is to establish baselines by evaluating existing models on this data, thereby transparently highlighting this problem and providing a reference point for future studies.

We concur with the reviewer that simply using Lovasz loss is only a partial mitigation, not a complete solution. However, we want to point out that **fully addressing this deep-rooted imbalance is a significant research challenge in its own right**, likely requiring multiple dedicated methodology papers focusing on novel algorithms (e.g., specialized loss functions, data resampling strategies, or generative augmentation). **Such an endeavor, while crucial, extends beyond the scope of a single dataset-focused paper**. Our work aims to provide a foundational dataset to enable and inspire that future research. This is precisely why we highlight this issue in our manuscript: to serve not only as a caution to users but also as a clear focus for future methodological advancements.

Thank you again for providing us with the opportunity to clarify the context and scope of our contribution.

**Q3:** The discussion on cross-disaster generalization needs to be strengthened. Table 6 shows that in different disaster types, certain events perform particularly poorly. In particular, the mIoU values are the lowest for explosion and chemical accident events such as Bata-EP-2021 and Kyaukpyu-CC-2023, while earthquake events such as Morocco-EQ-2023 and Noto-EQ-2024 also remain highly challenging. This indicates that the models face significant difficulties in handling highly destructive, structurally complex, and spatially heterogeneous disaster scenarios. The authors should analyze these challenges in more depth, such as the heterogeneity and extreme local variations in explosion damage, the diversity of collapse patterns in earthquake events, and the limitations of SAR data in capturing fine-grained details. It is also recommended to provide typical error cases and compare model errors across disaster types to better illustrate the shortcomings in generalization.

**R3:** We sincerely thank the reviewer for these detailed and highly constructive suggestions. The points raised are crucial for understanding the complexities of cross-disaster generalization, and we appreciate the opportunity to clarify and strengthen our discussion.

Our response to this valuable comment is structured in three parts: 1) we will gently clarify the context of the tables to ensure a common understanding of the results; 2) we will guide the reviewer to the sections in our manuscript where we already performed the in-depth analysis you suggested; 3) we will describe the new content we have added based on your excellent recommendation.

First, we would like to gently clarify this point to avoid any potential misunderstanding regarding the tables. **The results in Table 6 are part of the standard machine learning evaluation**, providing an

event-wise breakdown of the overall results shown in Table 5. The purpose of Table 6 is to prevent the evaluation from being dominated by events with a large number of samples, thus offering a more granular view of model performance on each disaster event. The experiments specifically designed to evaluate cross-event transfer generalization are presented and discussed in Section 5.4 (Section 4.6 in the revised manuscript), with the quantitative results shown in Table 8 (Table 10 in the revised manuscript). We apologize if this structure was not sufficiently clear.

Secondly, we are pleased that the reviewer highlighted these critical areas for analysis, as we also believe they are central to understanding the challenges. We would like to respectfully guide the reviewer to the following sections of our manuscript where **some of these points were discussed** in detail:

➢ Regarding the heterogeneity of damage, we analyzed this extensively in [Section 5.4.3: Why is cross-event transfer challenging] (Section 4.6.3 in the revised manuscript). Specifically, **Figure 9 provides violin plots** that visualize the significant shifts in SAR backscatter distributions for damaged and destroyed buildings across different events, including those of the same disaster type. This directly addresses the "heterogeneity and extreme local variations" and "diversity of collapse patterns" that the reviewer mentioned.

➢ Regarding model performance across disaster types and SAR limitations: This was analyzed in [Section 5.2 - What have the models learned and what can they learn] (Section 4.2 in the revised manuscript). The **bar chart in Figure 8** directly compares the models' average IoU across seven major disaster types. The accompanying text discusses the varying performance, explicitly noting the models' accuracy/errors across disaster types.

Finally, we **completely agree with the reviewer that providing typical error cases is an excellent way** to visually illustrate the model's generalization shortcomings. While our original analysis was primarily quantitative, visual examples provide a more intuitive understanding of the specific failure modes. Therefore, we have now added **a new figure (Figure 9 in the revised manuscript) and corresponding analysis to Section 4.2** in the revised manuscript. This new content shows concrete examples of model misclassifications. We believe this addition, prompted by the reviewer's valuable suggestion, significantly strengthens our analysis by bridging the quantitative results with qualitative, real-world examples of model errors.

Here, we show the revised part below for your convenience.

[Figure]

**Figure 9.** Typical failure cases of different models on Bata-Explosion-2021 and Noto-Earthquake-2024 in BRIGHT.

heterogeneous patterns of structural collapse typical of seismic events, where damage is often subtle, partial, and highly variable. These conditions pose significant challenges for SAR-based assessment. Interestingly, the model achieves relatively high IoU scores in the flood and hurricane events for the "Damaged" category with approximately 50% and 60%, respectively. This indicates that SAR effectively captures contextual environmental changes, such as water inundation or terrain disruption, which indirectly aid in assessing building damage. In the case of the conflict event, the model's performance on the "Destroyed" class is surprisingly low. This might be attributed to the limited number of destroyed samples in the dataset for this category, which leads to insufficient learning and poor generalization.

These quantitative limitations are vividly illustrated by the typical failure cases shown in Figure 9. In the Bata-Explosion-2021 event, models misclassify severely destroyed buildings as intact, reflecting the difficulty of interpreting heterogeneous debris patterns. Similarly, in the Noto-Earthquake-2024 event, large-scale collapses are largely missed, highlighting the challenge of diverse and subtle seismic damage. These examples visually confirm that while models can leverage broad contextual cues, they still struggle to distinguish partial from complete collapse where SAR backscatter changes are weak or inconsistent.

In summary, these findings confirm both the promise and limitations of optical-SAR modality for all-weather, global-scale disaster response. Although this combination performs well in events characterized by large-scale surface disruption (e.g., wildfires, volcanoes), it struggles with subtle or localized damage patterns. Incorporating richer data sources, such as fully polarimetric SAR and LiDAR data, can further enhance the accuracy and reliability of future all-weather building damage assessments.

**Q4:** The manuscript currently lacks a comparison with optical-only baselines, which is crucial to highlight the value of multimodal methods. Readers may question whether the inclusion of SAR brings significant benefits and whether the additional cost of multimodality is justified. To avoid such doubts, I suggest adding experiments with optical-only inputs and comparing them with Optical+SAR results. This would further emphasize the unique value of the BRIGHT dataset and provide stronger evidence for the necessity of multimodal fusion.

**R4:** We sincerely thank the reviewer for this excellent suggestion. The question of how multimodal

methods compare against an optical-only baseline is fundamental to justifying their value, and we appreciate the opportunity to provide a detailed clarification and new experimental evidence.

First, we would like to respectfully clarify the specific scenario that the BRIGHT dataset is designed to model. The core premise of BRIGHT is to facilitate **all-weather, rapid disaster response**. It is constructed around the common and challenging real-world situation where a disaster event (e.g., a hurricane, flood, or wildfire) is followed by adverse atmospheric conditions (e.g., cloud cover, smoke) that prevent the timely acquisition of usable post-event optical imagery. Therefore, **the dataset's composition is intentionally pre-event optical + post-event SAR**. This represents a pragmatic and operationally vital workflow. Consequently, a direct **"optical-only" baseline** you suggested (i.e., pre-event optical + post-event optical) **is not feasible on the main BRIGHT dataset** by design, as high-quality post-event optical imagery is not a component of its primary structure.

However, we completely agree with the reviewer that a direct comparison is crucial for understanding the relative strengths of each modality when both happen to be available under ideal conditions. To address this valuable point, we have **conducted a new set of experiments on a specific subset of our data**, including Bata-Explosion-2020, Beirut-Explosion-2021, Hawaii-Wildfire-2023, Libya-Flood-2023 and Noto-Earthquake-2024, for which high-quality, cloud-free and preprocessed post-event optical imagery was also available to us.

We benchmarked three different setups on this subset:

➢ Optical-Only: Pre-event optical + Post-event optical

➢ SAR-Only (**BRIGHT's standard**): Pre-event optical + Post-event SAR

➢ Optical+SAR Fusion: Pre-event optical + Post-event optical + Post-event SAR

As the results show (**Table 8** in the revised manuscript), when high-quality, cloud-free post-event optical imagery is available, the post-event optical approach outperforms the post-event SAR approach across all tested models. For instance, using the state-of-the-art **DamageFormer** model, the **post-event optical setup achieves a final mIoU of 69.76%,** higher than the **65.56% from the post-event SAR setup**. This is expected, as optical imagery provides rich and intuitive visual information for damage assessment. Crucially, the results also demonstrate that fusing both modalities consistently provides the best results, outperforming even the strong post-event optical methods in every case. For example, DamageFormer's mIoU increases from 69.76% (post-event optical) to **70.79% with the addition of SAR data**, suggesting that SAR provides complementary information that can enhance the results even when high-quality optical data is present.

While this experiment provides a valuable benchmark, its results ultimately reinforce our motivation for BRIGHT. The high performance of the **optical-only model is entirely contingent on the availability of ideal, cloud-free post-event imagery**, a condition frequently not met in the critical window after many disasters. Therefore, **the small performance trade-off of the post-event SAR-based approach is justified by its invaluable all-weather, day-and-night operational capability**. BRIGHT is designed precisely to advance the development of models for these realistic, often non-ideal, but operationally critical scenarios. We have added this new experiment and discussion to the new Section in the revised

manuscript to further emphasize the unique value of our dataset and the necessity of multimodal fusion.

Here, we show the revised part below for your convenience.

**Table 8.** Performance comparison of different post-event modalities on a subset of BRIGHT. Results are reported for UNet, DeepLabV3+, and DamageFormer on five disaster events where high-quality post-event optical imagery is available: Bata-Explosion-2020, Beirut-Explosion-2021, Hawaii-Wildfire-2023, Libya-Flood-2023, and Noto-Earthquake-2024.

| Method | Post-event modality | $F_1^{loc}$ (%) | $F_1^{clf}$ (%) | Final mIoU (%) | IoU per class (%) | | | |
|---|---|---|---|---|---|---|---|---|
| | | | | | Background | Intact | Damaged | Destroyed |
| UNet | SAR | 85.05 | 71.43 | 62.94 | 94.39 | 65.60 | 42.34 | 49.43 |
| | Optical | 86.46 | 75.64 | 65.96 | 94.56 | 68.62 | 45.27 | 55.36 |
| | Optical+SAR | 86.70 | 74.46 | 66.29 | 94.72 | 69.16 | 41.73 | 59.57 |
| DeepLabV3+ | SAR | 83.55 | 67.52 | 60.57 | 93.86 | 65.62 | 35.64 | 47.16 |
| | Optical | 85.79 | 74.39 | 64.87 | 94.33 | 67.90 | 44.31 | 52.94 |
| | Optical+SAR | 85.90 | 74.87 | 65.84 | 94.48 | 69.60 | 44.68 | 54.60 |
| DamageFormer | SAR | 88.41 | 73.43 | 65.56 | 95.30 | 70.62 | 41.31 | 55.00 |
| | Optical | 88.32 | 78.04 | 69.76 | 95.37 | 72.72 | 47.26 | 63.68 |
| | Optical+SAR | 88.86 | 79.27 | 70.79 | 95.56 | 73.64 | 48.44 | 65.51 |

**4.4 Impact of post-event modality on building damage assessment performance**

Although the primary design of BRIGHT is to facilitate all-weather disaster response through the use of pre-event optical and post-event SAR imagery, it is also important to understand how these modalities compare when high-quality post-event optical imagery is available. To this end, we conducted supplementary experiments on a subset of events, including Bata-Explosion-2020, Beirut-Explosion-2021, Hawaii-Wildfire-2023, Libya-Flood-2023, and Noto-Earthquake-2024, for which pre-processed post-event optical data were accessible. We evaluated three experimental setups: (i) optical-only (pre-event optical + post-event optical), (ii) SAR-only (pre-event optical + post-event SAR, i.e., the standard BRIGHT setting), and (iii) optical+SAR fusion (pre-event optical + post-event optical + post-event SAR).

Table 8 presents the experimental results. As expected, when ideal post-event optical imagery is available, the optical-only setup achieves higher performance than the SAR-only setup. For example, with DamageFormer, the optical-only configuration reaches a final mIoU of 69.76%, compared to 65.56% for SAR-only. Importantly, the performance gap between optical and SAR is not substantial, demonstrating that SAR alone provides a strong alternative in the absence of usable optical imagery. Moreover, the fusion of optical and SAR consistently yields the best results across all tested models. For instance, DamageFormer's mIoU further increases to 70.79% with Optical+SAR fusion, indicating that SAR contributes complementary information that strengthens performance even under optimal optical conditions.

These findings underscore two important insights. First, multimodal fusion is beneficial even when high-quality optical data are available, as SAR provides unique structural information that enriches the optical signal. Second, the performance of the SAR-only approach, being reasonably close to the optical-only results, highlights the practical value of SAR in real-world disaster scenarios where post-event optical imagery is often unavailable. BRIGHT is therefore designed to advance the development of models for these realistic, often non-ideal, but operationally critical all-weather disaster response settings.

**Q5:** The discussion of limitations and future directions is insufficient. At present, the conclusion mainly emphasizes the dataset's contributions, but it does not address its shortcomings in detail. It is suggested to include a separate subsection summarizing the limitations, such as the use of singlepolarization SAR, the lack of time-series data, and the fact that most disaster events are concentrated after 2020.

**R5:** We sincerely appreciate this constructive suggestion to improve the discussion on the limitations of our dataset. We fully agree that a thorough and transparent account of the dataset's shortcomings is essential for the community. Just for clarity, our original manuscript actually already included relevant content in Section 6 – Discussion:

➢ **Section 6.1 - Limitation of BRIGHT**: to address what we identified as the primary limitations, including potential registration errors, label quality, and sample/regional imbalance.

➢ **Section 6.2 - Significance of BRIGHT**: by suggesting that "Incorporating richer data sources, such as fully polarimetric SAR and LiDAR data, can further enhance the accuracy and reliability of future all-weather building damage assessments".

That said, we acknowledge that our initial discussion did not explicitly address two important points raised by the reviewer: the absence of time-series data and the temporal concentration of disaster events after 2020. We greatly appreciate this observation. In response, we have **revised Section 6.1 (Section 5.1 in the revised manuscript) to incorporate these limitations**, thereby providing a more comprehensive discussion. We believe these additions, prompted by the reviewer's insightful comment, have significantly strengthened the manuscript. We thank the reviewer again for helping us improve the quality of our work.

Here, we show the revised part below for your convenience.
* * *
**5  Discussion**

**5.1  Limitation of BRIGHT**

We begin this subsection by acknowledging that the composition of the BRIGHT dataset is fundamentally shaped by practical constraints in data availability. While BRIGHT represents a significant step forward in assembling a large-scale, multimodal, and globally distributed dataset for disaster response, it is important to recognize several inherent limitations. These limitations arise not only from the scarcity of open-access VHR SAR imagery, especially over disaster-affected regions, but also from the challenges of manual annotation and the uneven distribution of events. To provide a clearer picture for potential users, we summarize these constraints in four aspects below.

4. **Modality and temporal scope.** The dataset's scope is defined by two key characteristics of the available data. First, it exclusively utilizes single-polarization SAR imagery. The current version lacks the more informative multi-polarization or dense time-series SAR data, which, if available, could enable more nuanced damage characterization and long-term recovery monitoring, respectively. Second, the dataset's temporal coverage is concentrated on events from 2020 onwards. This is a direct consequence of its reliance on modern commercial VHR SAR providers (Capella Space and Umbra), whose open-data initiatives largely commenced around that time.
* * *
**Q6:** The description of study areas and disaster events is somewhat redundant.

**R6:** Thank you so much for this valuable suggestion. In preparing the manuscript, we followed the style

of other disaster-related papers published in ESSD, which typically provide detailed descriptions of study areas and events. However, we agree with you that presenting too many event-specific details in the main text can be overwhelming and may distract readers from the core contributions of the work.

In response, we have **revised the structure of the manuscript to streamline this section**. Specifically, we have **moved the detailed descriptions of individual disaster events to the Appendix**, while retaining a concise overview in the main text. Specifically, the general description originally included in Section 2 has been merged into Section 3, now serving as its opening subsection. We believe this restructuring improves the readability of the manuscript by highlighting the key information while still making the detailed event descriptions accessible to interested readers.

Here, we show the revised part below for your convenience.
* * *
**2  Dataset Description**

**2.1  Study areas and disaster events**

140   We selected 14 disaster events across the globe for BRIGHT, as illustrated in Figure 2 and Table 2. Since both Capella Space and Umbra satellites were launched in 2020, we focused on study areas where disasters have occurred since then. The selected regions are primarily in developing countries, where public administration and disaster response capacities tend to be weaker compared to those in developed nations, making international assistance more critical. The dataset covers five major types of natural disasters: earthquakes, storms (including hurricanes and cyclones), wildfires, floods, and volcanic eruptions. Addi-

145   tionally, it includes human-made disasters, such as accidental explosions and armed conflicts. Detailed descriptions of the 14 disaster events are provided in Appendix A.
* * *
**Q7:** The terminology for "one-shot" may not be accurate. The authors describe it as using "a small number of labeled samples," which may be better defined as "few-shot." Since these concepts are borrowed from previous work, it is suggested to cite the corresponding references.

**R7:** We sincerely thank you for this precise and helpful comment. We agree that a clear and accurate definition of terms like "one-shot" and "few-shot" is crucial for methodological rigor.

The reviewer is correct that the phrase "a small number of labeled samples" generally corresponds to a "few-shot" learning scenario. Our intention in **using this more general phrase** in the original manuscript was to illustrate the practical context of disaster response, where it might be feasible for experts to quickly label a handful of examples from a new event. We recognize that this phrasing created an ambiguity between the real-world analogy and our specific experimental setup. Therefore, we **have modified the corresponding contents** in the revised manuscript.

Furthermore, we would like to take this opportunity to clarify a key distinction in our application of the "one-shot" paradigm. In many computer vision tasks, one-shot learning typically refers to learning to recognize new semantic classes from a single example. In our work, however, the classes (e.g., intact, damaged, destroyed) remain consistent across all disaster events. Our challenge is not learning new

classes but rather adapting the model to a new data domain. That is a new, unseen disaster. Therefore, we use the one-shot sample to facilitate cross-event adaptation, helping the model adjust to the unique visual characteristics, sensor properties, and damage signatures of the target event.

To ensure our manuscript accurately reflects this, we have revised the description to be more precise about both the terminology and the methodological context. Here, we show the revised part below for your convenience.
* * *
225    • One-shot setup: Recognizing the difficulty of the zero-shot setup, we introduce a one-shot setup. This setting simulates a realistic scenario where a single, representative sample from the new disaster can be quickly labeled to guide model adaptation. In this setting, a limited subset of labeled data (one pair for training and one pair for validation) from the target disaster event is incorporated into the training process. At the same time, the majority of the test set remains unseen. This setup evaluates the model's ability to leverage a minimal amount of manually labeled data to improve
230    disaster-specific adaptation.

It is worth noting that our cross-event transfer setup differs from classic few-shot learning tasks in the computer vision field (Amirreza Shaban and Boots, 2017; Wang et al., 2020). Our goal is not to recognize new classes, but to adapt the model's knowledge of existing classes to a new domain, *i.e.*, an unseen disaster event.
* * *
**Q8:** At line 420, the authors state that SAR is not sensitive to fine structural changes. Would this limitation reduce the value of multimodality in certain scenarios?

**R8:** Thank you for your thoughtful comment. We thank the reviewer for raising this insightful question. It is important to clarify that **every remote sensing modality captures only certain aspects of the Earth's surface, and each has its own strengths and limitations**. For example, optical imagery records reflect light in the visible and near-infrared spectrum, while SAR measures backscattered microwave signals. Consequently, each modality is inherently more or less sensitive to particular types of features.

In the case of single-polarization SAR, it is true that very fine structural changes/damages may not be well captured. However, **this limitation is not unique to SAR**. Indeed, even optical VHR satellite imagery can sometimes struggle to detect subtle or small-scale damage, as noted in [1]. Therefore, the challenge of capturing fine-grained structural changes/damages is a broader limitation of satellite data rather than a drawback that renders multimodality less valuable.

The core value of multimodal integration lies in ensuring operational continuity for disaster response, which is the primary motivation for our dataset. As discussed in the Introduction, optical EO data, while semantically rich, is frequently rendered unusable by cloud cover in the critical hours and days following a disaster. SAR data's all-weather capability is not just an advantage; it is often the only viable option for timely data acquisition. **The combination of pre-event optical data and post-event SAR data is therefore a pragmatic and powerful solution for rapid assessment,** even if neither modality alone is perfect.

Of course, in an ideal scenario, incorporating additional modalities such as fully polarimetric SAR,

LiDAR, or multi-temporal imagery, would provide more complete coverage of structural damages. As noted in Section 6.2 (Section 5.2 in the revised manuscript), we explicitly highlight this as an important direction for future dataset development.

In summary, while we acknowledge the limitations of single-polarization SAR for detecting fine-scale damage, this represents **a pragmatic trade-off rather than a fundamental flaw**. It does not diminish the value of multimodality but instead highlights the importance of combining complementary data sources to build robust, timely, and practical disaster response systems.

[1] T. Manzini, P. Perali, J. Tripathi and R. R. Murphy, "Now you see it, Now you don't: Damage Label Agreement in Drone & Satellite Post-Disaster Imagery," *Proc. 2025 ACM Conf. Fairness, Accountability, and Transparency (FAccT '25)*, New York, NY, USA, pp. 1998–2008, 2025.

---

## Author Comment (AC5)

**Response to Comments of Referee #1**

Thank you for the instructive and constructive comments for our paper. Those comments are very helpful for and serve as significant guidance for our research. We have studied the comments carefully and revised our manuscript accordingly. The changes in our manuscript are highlighted in **red**. The point-to-point responses to your questions/comments are listed as follows.

**Comments to the Author:**

This paper introduces BRIGHT, a novel and timely benchmark dataset for building damage assessment using multimodal high-resolution optical and SAR imagery. Covering 14 globally distributed disaster events, BRIGHT provides pixel-level damage annotations for over 384,000 buildings. The dataset is designed to facilitate AI-based disaster response research, particularly in challenging all-weather conditions. The authors also benchmark a suite of machine learning and deep learning models on multiple tasks. The authors provided detailed documents and descriptions, making the data, related source code, and pretrained weights of models easy to understand and use.

In summary, this is quite interesting and solid work. I'd like to recommend the acceptance of this work since it represents an important contribution to Earth observation and disaster response communities. Yet before acceptance, several clarifications and refinements are suggested.

**Response:** We really appreciate your spot-on summary of our manuscript and such a positive endorsement of our work. Our responses to your valuable comments and suggestions are itemized below.

**Q1:** The manuscript would benefit from deeper exploration of what the models learn from multimodal fusion. Specifically, what roles do optical images play in multimodal building damage assessment? Is it beyond just building footprint localization? On the other words, are the features extracted from optical imagery actively compared with SAR representations? Some discussion (e.g., based on CAMs in Fig. 7) is provided but can be more explicitly elaborated.

**R1:** Thank you so much for this very insightful comment. To investigate the role of optical imagery in multimodal building damage assessment, we conducted additional experiments as suggested. Specifically, we evaluated UNet and DeepLabV3+ under two input settings: optical + SAR and SAR only. We chose these two models because they are a single-branch architecture, making it **straightforward to adjust the number of input channels by modifying the first convolutional layer**. In contrast, the other five methods adopt Siamese networks, where structural changes for different input modalities would require extensive reconfiguration. For UNet and DeepLabV3+, the modification introduces

negligible changes in the parameter count. To **isolate the contribution of optical imagery** beyond building footprint localization, we **provided all models with perfect building masks** as post-processing steps prior to evaluation.

The results, presented in **Table 7** of the revised manuscript, demonstrate that optical imagery contributes significantly to distinguishing different damage levels. When provided with **optical + SAR inputs**, both models show **notable improvements in the IoU scores for the "Damaged" and "Destroyed"** classes compared to SAR-only inputs. For example, UNet's IoU for "Damaged" improved from 35.83% to 44.83%. DeepLabV3+ also benefits from optical imagery, with IoU for "Damaged" changing from 39.63% (SAR only) to 40.45% (optical + SAR), and for "Destroyed" increasing substantially from 59.54% to 64.94%. These findings indicate that **optical imagery provides critical complementary information** that supports damage classification, rather than merely improving building localization.

Accompanying Table 7, we have added a new section (**Section 4.3** in the revised manuscript) to the revised manuscript to provide a more detailed discussion of these findings. We show the revised part below for your convenience.

**Table 7.** Performance comparison of UNet and DeepLabV3+ using only post-event SAR and pre-event optical plus post-event SAR inputs for damage classification task. Here, accurate building masks are provided as the post-processing step to all models to isolate the effect of building localization on the damage classification task.

| Method | Modality | $F_1^{clf}$ (%) | Final mIoU (%) | IoU per class (%) | | | |
|---|---|---|---|---|---|---|---|
| | | | | Background | Intact | Damaged | Destroyed |
| UNet | Post-event SAR | 68.71 | 69.84 | 100.0 | 88.19 | 35.83 | 55.35 |
| | Pre-event optical + post-event SAR | 73.59 | 72.41 | 100.0 | 89.38 | 44.83 | 55.42 |
| DeepLabV3+ | Post-event SAR | 72.12 | 72.19 | 100.0 | 89.59 | 39.63 | 59.54 |
| | Pre-event optical + post-event SAR | 73.90 | 73.93 | 100.0 | 90.32 | 40.45 | 64.94 |

polarimetric SAR and LiDAR data, can further enhance the accuracy and reliability of future all-weather building damage assessments.

**4.3 The role of optical pre-event data in multimodal building damage assessment**

370  In the last section, CAM visualizations revealed that DL models also exhibit responses to disaster-specific patterns in pre-event optical imagery. This observation suggests that optical data may play a more complex role in multimodal building damage mapping than simply supporting building localization. In other words, in a multimodal bi-temporal setup, does pre-event optical imagery act solely as a localization aid, or does it provide additional semantic cues that networks can exploit for more accurate damage classification?

375  To explore this, we conducted controlled experiments using UNet and DeepLabV3+. Both networks were trained under two configurations: (i) using post-event SAR imagery only, and (ii) using multimodal pre- and post-event inputs (optical-SAR). To isolate the contribution of pre-event optical data beyond building localization, we provided perfect building masks for postprocessing in both settings. This design ensures that any observed differences in performance are attributable to the additional information from pre-event optical imagery, rather than differences in network architecture or localization accuracy.

380  The results, summarized in Table 8, show that incorporating pre-event optical imagery leads to notable improvements in distinguishing building damage levels. For UNet, the IoU for the "Damaged" class increased from 35.83% (SAR only) to 44.83% (Optical-SAR), and for the "Destroyed" class from 55.35% to 55.42%. DeepLabV3+ exhibited significant gains also, with IoU improvements from 39.63% to 40.45% for "Damaged" category, and from 59.54% to 64.94% for "Destroyed" category. These results suggest that pre-event optical imagery contributes beyond mere building localization, enriching the feature space for

385  more effective semantic comparison for different building damage levels across modalities.

**Q2:** The manuscript makes extensive evaluations of supervised and unsupervised change detection models, but the conceptual and methodological relationship between building damage assessment and generic change detection remains unclear, which is largely implied rather than discussed. An explicit and clearer explanation would be great for readers who lack of related background.

**R2:** Thank you for this insightful comment. We agree that clarifying the conceptual and methodological relationship between building damage assessment (BDA) and generic change detection (CD) will help readers unfamiliar with the field.

Specifically, a common view is to treat **BDA as a special case of "one-to-many" semantic change detection** tasks [1]-[4], where the goal is to assess not just whether a change has occurred but also to characterize the type and severity of the change (*i.e.*, levels of damage). In this sense, BDA extends beyond binary change detection by requiring finer-grained semantic interpretation of pre- and post-event imagery. **Many existing methods for BDA are thus derived from or adapted versions of generic change detection models**. Furthermore, in some unified change detection frameworks [3]-[5], **BDA is explicitly included as one of the downstream tasks**, highlighting their methodological overlap.

It is important to note that this discussion focuses on the formulation of BDA tasks that take bi-temporal inputs (i.e., both pre- and post-disaster images). Alternative approaches that rely solely on post-disaster imagery exist but are outside the scope of our evaluation and discussion.

We have added the above description in original Section 4.1 (Section 3.1 in the revised manuscript) to clarify this problem. We show the revised part below for your convenience.

[1] Z. Zheng, Y. Zhong, J. Wang, A. Ma, and L. Zhang, "Building damage assessment for rapid disaster response with a deep object-based semantic change detection framework: From natural disasters to man-made disasters," *Remote Sensing of Environment*, vol. 265, p. 112636, 2021.

[2] W. Lu, L. Wei and M. Nguyen, "Bitemporal Attention Transformer for Building Change Detection and Building Damage Assessment," *IEEE Journal of Selected Topics in Applied Earth Observations and Remote Sensing*, vol. 17, pp. 4917-4935, 2024.

[3] H. Chen, J. Song, C. Han, J. Xia and N. Yokoya, "ChangeMamba: Remote Sensing Change Detection with Spatiotemporal State Space Model," *IEEE Transactions on Geoscience and Remote Sensing*, vol. 62, pp. 1-20, 2024.

[4] Z. Zheng, Y. Zhong, J. Zhao, A. Ma, and L. Zhang, "Unifying remote sensing change detection via deep probabilistic change models: From principles, models to applications," *ISPRS Journal of Photogrammetry Remote Sensing*, vol. 215, pp. 239–255, 2024.

[5] H. Guo, X. Su, C. Wu, B. Du and L. Zhang, "SAAN: Similarity-Aware Attention Flow Network for Change Detection with VHR Remote Sensing Images," *IEEE Transactions on Image Processing*, vol. 33, pp. 2599-2613, 2024.

It is worth noting that in this work, we focus on the formulation of building damage assessment as a bi-temporal task, where both pre- and post-event images are used as inputs. This formulation aligns closely with generic change detection tasks, which aim to identify changes between two time points. Conceptually, building damage assessment can be viewed as a specialized "one-to-many" semantic change detection problem (Zheng et al., 2021, 2024; Lu et al., 2024), where the objective is not only to detect whether a change has occurred but also to categorize the type and severity of changes (damages) to buildings. Many existing methods are thus derived from or adapted versions of generic change detection frameworks (Chen et al., 2024; Zheng et al., 2024; Guo et al., 2024).

**3.2  Benchmark suites**

**Q3:** Since UMCD methods underperform, consider including a random guessing baseline for reference. This would contextualize the difficulty of BRIGHT and help readers understand the performance floor under UMCD setup.

**R3:** Thank you for your insightful suggestion. We have added the results of a random guessing baseline to Table 11 **(now Table 13 in the revised manuscript)** for reference. As shown below, the different methods achieve improvements over random guessing; however, the gains are not very significant. This highlights the challenging nature of applying UMCD methods to the BRIGHT dataset. We show the revised part below for your convenience.

**Table 13.** Results of representative unsupervised multimodal change detection methods. KC is the acronym of kappa coefficient. The highest values are highlighted in **bold**, and the second-highest results are highlighted in underline. The accuracies on the UMCD benchmark dataset are the accuracies on the four datasets presented in Figure H1, obtained from their literature. Details of methods and benchmark datasets are presented in Appendix H. The random guessing baseline is included to indicate the performance floor under the UMCD setup. The "-" symbol indicates that the corresponding method did not report results on that dataset in their original publications.

| Method | UMCD benchmark datasets | | | BRIGHT | | | |
|---|---|---|---|---|---|---|---|
| | OA | F1 | KC | OA | F1 | IoU | KC |
| *Random guessing* | 50.0 | 8.4 / 6.0 / 11.0 / 11.4 | 0.0 | 50.00 | 7.83 | 4.08 | 0.00 |
| IRG-McS (Sun et al., 2021) | 98.3 / - / 97.1 / 97.2 | 80.4 / - / 75.4 / 73.7 | 79.4 / - / 73.9 / 75.1 | **90.03** | 12.65 | 6.75 | 7.65 |
| SR-GCAE (Chen et al., 2022b) | 98.6 / 98.5 / - / - | 82.9 / 77.6 / - / - | 82.1 / 76.9 / - / - | 77.83 | 14.35 | 7.73 | 5.64 |
| FD-MCD (Chen et al., 2023) | 98.2 / 97.8 / - / 96.7 | 81.4 / 72.2 / - / 73.2 | 82.3 / 71.1 / - / 71.4 | 80.96 | **15.84** | **8.60** | 7.94 |
| AOSG (Han et al., 2024) | - / - / - / 96.4 | - / - / - / 77.7 | - / - / - / 75.9 | 77.93 | 10.75 | 5.68 | 3.98 |
| AGSCC (Sun et al., 2024a) | 98.3 / - / 95.9 / 97.6 | 78.2 / - / 68.0 / 77.9 | 77.3 / - / 65.8 / 76.6 | 88.49 | 14.82 | 8.00 | **9.54** |
| AEKAN (Liu et al., 2025) | 98.7 / - / - / 98.3 | 83.8 / - / - / 84.7 | 83.1/ - / - / 83.9 | 81.60 | 13.09 | 7.00 | 3.56 |

**Q4:** While Table 1 offers a comprehensive comparison of datasets, several datasets seem relevant and should be included to enhance its completeness, like CRASAR-U-DROIDs [arXiv:2407.17673] and Noto-Earthquake building damage dataset [10.5194/essd-2024-363].

**R4:** Thank you for your valuable suggestion. We have reviewed the CRASAR-U-DROIDs [arXiv:2407.17673] and the Noto-Earthquake Building Damage Dataset [10.5194/essd-2024-363] and have updated **Table 1** to include them for a more comprehensive comparison.

We show the corresponding revised part below for your convenience.

**Table 1.** Comparison of BRIGHT with the existing building damage assessment datasets. The OA indicates whether the dataset is open access (OA) or not, and GSD is an acronym for ground sampling distance (GSD). Note that since some datasets integrate other datasets, we summarize only the largest one to avoid duplication here. For example, the BDD dataset (Adriano et al., 2021) includes the Tohoku-Earthquake-2011 dataset (Bai et al., 2018) and Palu-Tsunami-2018 dataset (Adriano et al., 2019).

| Dataset | OA | Modality | GSD (m/pixel) | No. of events | Disaster type | No. of building | Granularity |
|---|---|---|---|---|---|---|---|
| ABCD (Fujita et al., 2017) | ✓ | Optical EO | 0.4 | 1 | Tsunami | N/A | Image-level |
| (Nguyen et al., 2017) | ✓ | Images on social media | N/A | 4 | 3 natural disasters | N/A | Image-level |
| (Cheng et al., 2021) | ✓ | Optical EO | N/A | 1 | Hurricane | 1,802 | Image-level |
| (Xue et al., 2024) | ✓ | Street-view image | N/A | 1 | Hurricane | 2,468 | Image-level |
| FloodNet (Rahnemoonfar et al., 2021) | ✓ | Optical EO | N/A | 1 | Flood | 6,675 | Pixel-level |
| RescueNet (Rahnemoonfar et al., 2023) | ✓ | Optical EO | N/A | 1 | Hurricane | 10,903 | Pixel-level |
| Ida-BD (Kaur et al., 2023) | ✗ | Optical EO | 0.5 | 1 | Hurricane | 18,083 | Pixel-level |
| CRASAR-U-DROIDs (Manzini et al., 2024) | ✓ | Optical EO | 0.02-0.12 | 10 | 4 natural disasters 1 man-made disaster | 21,716 | Pixel-level |
| Noto-BDA-MV (Vescovo et al., 2025) | ✓ | Optical EO | N/A | 1 | Earthquake | 140,208 | Pixel-level |
| xBD (Gupta et al., 2019) | ✓ | Optical EO | <0.8 | 15 | 6 natural disasters | >700,000 | Pixel-level |
| QQB (Sun et al., 2024b) | ✓ | Optical and SAR EO | <1 | 1 | Earthquake | 4,029 | Pixel-level |
| BDD (Adriano et al., 2021) | ✗ | Optical and SAR EO | 1.2-3.3 | 9 | 3 natural disasters | 123,453 | Pixel-level |
| BRIGHT | ✓ | Optical and SAR EO | 0.3-1 | 14 | 5 natural disasters 2 man-made disasters | 384,596 | Pixel-level |

**Q5:** The paper describes careful multimodal alignment but omits the software used, e.g., ENVI, ArcGIS, or QGIS. Please provide related details.

**R5:** Thank you for mentioning this detail. We have added information about the multimodal registration process in Appendix B (**Appendix C in the revised manuscript**). Specifically, we used QGIS as the registration software, employing the [Georeferencer] plugin to align SAR images to the optical imagery as the reference. The transformation type was set to [Thin Plate Spline], and [Lanczos resampling (6×6 kernels)] was applied to ensure high-quality interpolation.

We show the corresponding revised part below for your convenience.

**Appendix C: Manual registration and estimating registration errors**

We performed the manual registration process using QGIS, with the "Georeferencer" plugin to align SAR images to the optical imagery as the reference. The transformation type was set to "Thin Plate Spline", and "Lanczos resampling (6×6 kernels)" was applied to achieve high-quality interpolation. The manually selected control points by EO experts on some disaster scenes are shown in Figure C1.

750

**Q6:** Appendix G includes important new experimental setups and evaluation methods for UMCD. However, too much content is composed together now. It is not easy for people to grasp information. Adding section subtitles could improve readability.

**R6:** Thank you for this helpful suggestion. According to your suggestion, we have revised Appendix G by dividing it into two parts for improved clarity. The first part introduces the unsupervised multimodal change detection methods, while the second part describes the proposed more practical evaluation protocol. This restructuring makes it easier for readers to grasp the key information.

**Q7:** 8: Please specify in the figure or caption that the values represent average ± standard deviation across models.

**R7:** Thank you for your thoughtful comment. We believe you were referring to **Figure 8**. We have clarified in the caption that each bar represents the mean IoU of seven deep learning models for a specific class under each disaster type, and the error bars indicate the standard deviation of IoU scores across the seven models.

We show the corresponding revised part below for your convenience.

[Figure]

**Figure 8.** IoU distribution of deep models over seven disaster types. Each bar represents the average IoU of seven DL models for that specific category under each disaster type. The error bars indicate the standard deviation of IoU scores across the seven models.

**Q8:** 10: Add a note in the caption to clarify that each dot corresponds to performance on a single test event under cross-event transfer.

**R8:** Thank you for your thoughtful comment. We believe you were referring to **Figure 10 (Figure 11 in**

**the revised manuscript)**. We have added a note in the caption to clarify that each dot represents the performance on a single test event under cross-event transfer.

We show the corresponding revised part below for your convenience.

[Figure]

**Figure 11.** Comparison of models' best performance (mIoU) on test events versus the best checkpoints selected on validation sets under the cross-event transfer setting. Each point represents the performance on a single test event under cross-event transfer. The farther a point lies from the diagonal line, the larger the gap between the model's selected performance and its true upper bound.

**Q9:** Typo in Table 7: "Object-based major voting" should be corrected to "Object-based majority voting".

**R9:** Thank you for your careful review. We have corrected "Object-based major voting" to "Object-based majority voting" in Table 7 (now **Table 9** in the revised manuscript).

We show the corresponding revised part below for your convenience.

**Table 9.** Further contributions to mIoU from post-processing algorithms. ChangeMamba (Chen et al., 2024) is used here as the baseline. Details on these algorithms are provided in Appendix E.

| Method | mIoU (set) | mIoU (event) |
|---|---|---|
| Baseline | 67.63 | 51.39 |
| Test-time augmentation | 68.50 | 51.95 |
| Object-based majority voting | 67.22 | 52.08 |
| Ensembling multiple models | 68.45 | 52.14 |
| All | 68.86 | 52.31 |

**Q10:** Clarify the meaning of "−" symbols in Table 11. Do they indicate missing data or inapplicability? This should be stated explicitly.

**R10:** Thank you for pointing this out. The "−" symbols in Table 11 indicate that the corresponding methods did not report results on that dataset in their original publications. We have clarified this in the caption of Table 11 (now **Table 13** in the revised manuscript). We show the corresponding revised part below for your convenience.

**Table 13.** Results of representative unsupervised multimodal change detection methods. KC is the acronym of kappa coefficient. The highest values are highlighted in **bold**, and the second-highest results are highlighted in underline. The accuracies on the UMCD benchmark dataset are the accuracies on the four datasets presented in Figure H1, obtained from their literature. Details of methods and benchmark datasets are presented in Appendix H. The random guessing baseline is included to indicate the performance floor under the UMCD setup. The "-" symbol indicates that the corresponding method did not report results on that dataset in their original publications.

| Method | UMCD benchmark datasets | | | BRIGHT | | | |
| --- | --- | --- | --- | --- | --- | --- | --- |
| | OA | F1 | KC | OA | F1 | IoU | KC |
| *Random guessing* | 50.0 | 8.4 / 6.0 / 11.0 / 11.4 | 0.0 | 50.00 | 7.83 | 4.08 | 0.00 |
| IRG-McS (Sun et al., 2021) | 98.3 / - / 97.1 / 97.2 | 80.4 / - / 75.4 / 73.7 | 79.4 / - / 73.9 / 75.1 | **90.03** | 12.65 | 6.75 | 7.65 |
| SR-GCAE (Chen et al., 2022b) | 98.6 / 98.5 / - / - | 82.9 / 77.6 / - / - | 82.1 / 76.9 / - / - | 77.83 | 14.35 | 7.73 | 5.64 |
| FD-MCD (Chen et al., 2023) | 98.2 / 97.8 / - / 96.7 | 81.4 / 72.2 / - / 73.2 | 82.3 / 71.1 / - / 71.4 | 80.96 | **15.84** | **8.60** | 7.94 |
| AOSG (Han et al., 2024) | - / - / - / 96.4 | - / - / - / 77.7 | - / - / - / 75.9 | 77.93 | 10.75 | 5.68 | 3.98 |
| AGSCC (Sun et al., 2024a) | 98.3 / - / 95.9 / 97.6 | 78.2 / - / 68.0 / 77.9 | 77.3 / - / 65.8 / 76.6 | 88.49 | 14.82 | 8.00 | **9.54** |
| AEKAN (Liu et al., 2025) | 98.7 / - / - / 98.3 | 83.8 / - / - / 84.7 | 83.1 / - / - / 83.9 | 81.60 | 13.09 | 7.00 | 3.56 |

**Q11:** "ML" should be defined on its first use and consistently used thereafter instead of alternating with [machine learning].

**R11:** Thank you for carefully checking this detail. We have defined "ML" (machine learning) at its first occurrence in the manuscript and have revised the text to ensure consistent use of the abbreviation thereafter.

**Q12:** Standardize currency formatting (e.g., USD vs. US$).

**R12:** Thank you for carefully noting this. We have standardized the currency formatting throughout the manuscript and now consistently use "USD" to avoid ambiguity.

**Q13:** Define abbreviations such as IGN and GSI when first mentioned as data providers.

**R13:** Thank you for kindly reminding us of this. We have defined the abbreviations "GSI" and "IGN" in the caption of **Table 2** in the revised manuscript. Specifically, GSI refers to The Geospatial Information Authority of Japan, and IGN refers to The Instituto Geográfico Nacional (National Geographic Institute) of Spain.

We show the corresponding revised part below for your convenience.

**Table 2.** Summary of basic information of the BRIGHT dataset with disaster events listed in chronological order. GSI refers to the Geospatial Information Authority of Japan, and IGN refers to the Instituto Geográfico Nacional (National Geographic Institute) of Spain.

| Disaster area | Type of disaster | Date | GSD (m/pixel) | Data provider / source | No. of tiles | No. of building |
|---|---|---|---|---|---|---|
| Beirut, Lebanon | Explosion (EP) | 04 Aug. 2020 | 1 | Maxar & Capella | 133 | 25,496 |
| Bata, Equatorial Guinea | Explosion (EP) | 07 Mar. 2021 | 0.5 | Maxar & Capella | 107 | 8,893 |
| Goma, DR Congo | Volcano eruption (VE) | 22 May 2021 | 0.33 | Maxar & Capella | 123 | 18,741 |
| Les Cayes, Haiti | Earthquake (EQ) | 14 Aug. 2021 | 0.48 | Maxar & Capella | 73 | 18,918 |
| La Palma, Spain | Volcano Eruption (VE) | 19 Sept. 2021 - 13 Dec. 2021 | 0.3-0.35 | IGN (Spain) & Capella | 933 | 30,239 |

**Q14:** The format of references should be standardized. Some of these entries use abbreviations for journals, while others have full titles.

**R14:** Thank you for your kind reminder. We have standardized all references in the revised manuscript. Journal names are now uniformly abbreviated according to the Journal Title Abbreviations by Caltech Library, as required by ESSD.

**Response to Comments of Referee #2**

Thank you for the instructive and constructive comments for our paper. Those comments are very helpful for and serve as significant guidance for our research. We have studied the comments carefully and revised our manuscript accordingly. The changes in our manuscript are highlighted in **red**. The point-to-point responses to your questions/comments are listed as follows.

**Comments to the Author:**

This manuscript introduces the BRIGHT dataset, which is the first open building damage assessment dataset with global coverage, multi-hazard scenarios, multimodal imagery (Optical and SAR), and sub-meter resolution. The paper systematically describes data collection, annotation, and quality control methods, and validates the dataset with multiple deep learning models, including cross-disaster transfer (zero-shot and one-shot), semi-supervised, and unsupervised approaches. The dataset demonstrates clear novelty and practical value, and it is of significant importance for advancing research and applications in disaster emergency response, remote sensing, and artificial intelligence. Generally, the paper is well structured, logically clear, with detailed results and strong value in terms of data sharing. Although the manuscript is rich in content, there are still details that require improvement, and I recommend appropriate revisions.

**Response:** We really appreciate your spot-on summary of our manuscript and such a positive endorsement of our work. Our responses to your valuable comments and suggestions are itemized below.

**Q1:** The explanation of annotation consistency and reliability remains insufficient. Although the authors state that the data annotations were obtained from multiple institutions such as Copernicus EMS, UNOSAT, and FEMA and then refined manually, there may be inconsistencies in how different institutions define "damaged" and "destroyed". This could affect the consistency of annotations across disaster scenarios. It is therefore necessary to further elaborate on the process of unifying annotations, provide more detail on the manual refinement procedures.

**R1:** We sincerely thank the reviewer for raising this crucial point. Ensuring annotation consistency across different data sources and disaster types is paramount for the reliability of BRIGHT as a benchmark dataset, and we appreciate the opportunity to elaborate on our rigorous unification and refinement process.

 Our approach was a multi-stage process designed specifically to address the potential inconsistencies the reviewer has identified:

First, we recognized that the source agencies, while conceptually aligned, use slightly different grading scales. To address this, we **established a single, standardized three-tier classification scheme for all events** in BRIGHT: Intact (1), Damaged (2), and Destroyed (3), with clear definitions provided in Table 3 of our manuscript. This scheme served as the universal target for all incoming annotations.

Secondly, the reviewer correctly notes that the exact terminology and number of damage tiers can differ between agencies. However, **their underlying definitions for EO-based damage assessment are conceptually consistent**. All agencies grade damage based on visually verifiable structural failure. This conceptual alignment provided a solid foundation for our initial, rule-based mapping. The "Destroyed" category was the most consistent. Labels such as "Destroyed", "Collapsed", or "Completely Damaged" from all sources were directly mapped to our Destroyed (3) class. For partial damage, we aggregated multiple intermediate tiers. Labels like "Severe Damage", "Major Damage", "Highly Damaged", or "Moderately Damaged" were all mapped to our single Damaged (2) class. This conservative aggregation ensures that our "Damaged" category represents significant, visually verifiable structural harm.

Recognizing that subtle inconsistencies could persist even after the rule-based mapping, the most critical stage of our process was a comprehensive manual review and refinement. This final, expert-led stage served as the ultimate guarantor of consistency, ensuring that every annotation conforms to our unified standard. This procedure, conducted using tools like Google Earth Pro, involved:

➢ Correction of Inconsistencies: Our experts meticulously compared pre- and post-disaster VHR optical imagery for each annotation point to identify and correct discrepancies between the source label and the visual evidence.

➢ Harmonization of Ambiguous Labels: We paid special attention to **ambiguous source labels**, such as "Possibly Damaged". In these cases, if clear structural damage was not evident upon visual inspection, we adopted a conservative approach and re-classified the building as "Intact" to ensure a high confidence "Damaged" class.

➢ Disaggregation of Area-Based Labels: Crucially, we **identified and re-processed all area-based damage annotations** (i.e., where an entire block or neighborhood was assigned a single damage category). Our team manually disaggregated these coarse labels, assigning a precise, building-wise (point-level) damage label to each individual structure within the area. This step was vital for ensuring instance-level consistency and granularity across the entire dataset.

Through this systematic process of standardization, mapping, and exhaustive expert-led refinement, we have made every effort to harmonize the annotations and ensure that the final labels in BRIGHT are as consistent and reliable as possible. We have now added these details to the manuscript to make our process more transparent.

Here, we show the revised part below for your convenience.

to ensure accuracy. Damage annotations were obtained from Copernicus Emergency Management Service[6], the United Nations Satellite Centre (UNOSAT) Emergency Mapping Products[7], and the Federal Emergency Management Agency (FEMA)[8]. These annotations were derived through visual interpretation of high-resolution optical imagery captured before and after the disasters by EO experts, supplemented by partial field visits. To harmonize these diverse annotations and ensure consistency across all 14 disaster events, we implemented a rigorous, multi-stage process. First, we established a single, standardized three-tier classification scheme, including Intact (with pixel value 1), Damaged (with pixel value 2), and Destroyed (with pixel value 3), with clear definitions provided in Table 3, drawing on the frameworks of FEMA's Damage Assessment Operations Manual, EMS-98, the BDD dataset (Adriano et al., 2021), and the xBD dataset (Gupta et al., 2019). While the source agencies' terminology can differ (e.g., "Severe Damage" vs. "Major Damage"), their underlying definitions for EO-based assessment are conceptually consistent. We leveraged this alignment for an initial rule-based mapping, where various intermediate damage tiers were conservatively aggregated into our single "Damaged" category. Second, our team of EO experts conducted a comprehensive manual verification and refinement of every annotation using multi-temporal VHR imagery on platforms like Google Earth Pro. This final stage served as the ultimate guarantor of consistency. We paid special attention to ambiguous source labels, such as "Possibly Damaged". Adopting a conservative approach, these were re-classified as "Intact" if clear structural damage was not evident, thereby ensuring a high-confidence "Damaged" class. We also manually disaggregated all area-based annotations (i.e., where an entire block was assigned a single category). We re-processed these to assign a precise, building-wise damage label to each individual structure, ensuring instance-level consistency and granularity across the entire dataset.

**Q2:** The treatment of class imbalance is not sufficient. Figure 5(d) shows that intact buildings account for over 80%, while destroyed buildings account for less than 7%. This severe imbalance directly affects the accuracy of recognizing destroyed classes. Although the authors employed the Lovasz loss function to partially alleviate the issue, this is still not enough to solve the problem. Is this imbalance one of the reasons for the relatively low performance of the subsequent experimental results?

**R2:** We thank the reviewer for this insightful question. The reviewer has astutely identified one of the most significant and inherent challenges in the task of automated building damage assessment.

First, we would like to clarify that this severe class imbalance is not a unique artifact of our dataset but rather **a common characteristic of real-world post-disaster data**. Disasters, even when severe, typically damage or destroy a minority of the buildings in an affected area. For instance, the widely used **xBD dataset** exhibits a similar long-tail distribution, with the "intact" class constituting the vast majority of labeled buildings **(approximately 75%)**. The imbalance in BRIGHT, as shown in Figure 5(d), therefore realistically reflects the sparse nature of catastrophic damage.

Second, we agree with the reviewer that this imbalance is indeed one of the reasons for the lower performance observed in our benchmark results. Unlike general land-cover mapping tasks, where classes tend to be more balanced, the **rarity and variability of damage signatures** make it especially difficult for models to learn robust and generalizable representations from a limited number of disaster events.

Then, we wish to clarify the primary scope of our work. As a dataset and benchmark paper, **our central contribution is to capture and present these real-world challenges** in multimodal building

damage mapping, including the severe class imbalance, to **provide a realistic and challenging testbed for the community**. Our objective is to establish baselines by evaluating existing models on this data, thereby transparently highlighting this problem and providing a reference point for future studies.

We concur with the reviewer that simply using Lovasz loss is only a partial mitigation, not a complete solution. However, we want to point out that **fully addressing this deep-rooted imbalance is a significant research challenge in its own right**, likely requiring multiple dedicated methodology papers focusing on novel algorithms (e.g., specialized loss functions, data resampling strategies, or generative augmentation). **Such an endeavor, while crucial, extends beyond the scope of a single dataset-focused paper**. Our work aims to provide a foundational dataset to enable and inspire that future research. This is precisely why we highlight this issue in our manuscript: to serve not only as a caution to users but also as a clear focus for future methodological advancements.

Thank you again for providing us with the opportunity to clarify the context and scope of our contribution.

**Q3:** The discussion on cross-disaster generalization needs to be strengthened. Table 6 shows that in different disaster types, certain events perform particularly poorly. In particular, the mIoU values are the lowest for explosion and chemical accident events such as Bata-EP-2021 and Kyaukpyu-CC-2023, while earthquake events such as Morocco-EQ-2023 and Noto-EQ-2024 also remain highly challenging. This indicates that the models face significant difficulties in handling highly destructive, structurally complex, and spatially heterogeneous disaster scenarios. The authors should analyze these challenges in more depth, such as the heterogeneity and extreme local variations in explosion damage, the diversity of collapse patterns in earthquake events, and the limitations of SAR data in capturing fine-grained details. It is also recommended to provide typical error cases and compare model errors across disaster types to better illustrate the shortcomings in generalization.

**R3:** We sincerely thank the reviewer for these detailed and highly constructive suggestions. The points raised are crucial for understanding the complexities of cross-disaster generalization, and we appreciate the opportunity to clarify and strengthen our discussion.

Our response to this valuable comment is structured in three parts: 1) we will gently clarify the context of the tables to ensure a common understanding of the results; 2) we will guide the reviewer to the sections in our manuscript where we already performed the in-depth analysis you suggested; 3) we will describe the new content we have added based on your excellent recommendation.

First, we would like to gently clarify this point to avoid any potential misunderstanding regarding the tables. **The results in Table 6 are part of the standard machine learning evaluation**, providing an event-wise breakdown of the overall results shown in Table 5. The purpose of Table 6 is to prevent the evaluation from being dominated by events with a large number of samples, thus offering a more granular view of model performance on each disaster event. The experiments specifically designed to

evaluate cross-event transfer generalization are presented and discussed in Section 5.4 (Section 4.6 in the revised manuscript), with the quantitative results shown in Table 8 (Table 10 in the revised manuscript). We apologize if this structure was not sufficiently clear.

Secondly, we are pleased that the reviewer highlighted these critical areas for analysis, as we also believe they are central to understanding the challenges. We would like to respectfully guide the reviewer to the following sections of our manuscript where **some of these points were discussed** in detail:

➢ Regarding the heterogeneity of damage, we analyzed this extensively in [Section 5.4.3: Why is cross-event transfer challenging] (Section 4.6.3 in the revised manuscript). Specifically, **Figure 9 provides violin plots** that visualize the significant shifts in SAR backscatter distributions for damaged and destroyed buildings across different events, including those of the same disaster type. This directly addresses the "heterogeneity and extreme local variations" and "diversity of collapse patterns" that the reviewer mentioned.

➢ Regarding model performance across disaster types and SAR limitations: This was analyzed in [Section 5.2 - What have the models learned and what can they learn] (Section 4.2 in the revised manuscript). The **bar chart in Figure 8** directly compares the models' average IoU across seven major disaster types. The accompanying text discusses the varying performance, explicitly noting the models' accuracy/errors across disaster types.

Finally, we **completely agree with the reviewer that providing typical error cases is an excellent way** to visually illustrate the model's generalization shortcomings. While our original analysis was primarily quantitative, visual examples provide a more intuitive understanding of the specific failure modes. Therefore, we have now added **a new figure (Figure 9 in the revised manuscript) and corresponding analysis to Section 4.2** in the revised manuscript. This new content shows concrete examples of model misclassifications. We believe this addition, prompted by the reviewer's valuable suggestion, significantly strengthens our analysis by bridging the quantitative results with qualitative, real-world examples of model errors.

Here, we show the revised part below for your convenience.

These quantitative limitations are vividly illustrated by the typical failure cases shown in Figure 9. In the Bata-Explosion-2021 event, models misclassify severely destroyed buildings as intact, reflecting the difficulty of interpreting heterogeneous debris patterns. Similarly, in the Noto-Earthquake-2024 event, large-scale collapses are largely missed, highlighting the challenge of diverse and subtle seismic damage. These examples visually confirm that the significant heterogeneity in damage patterns makes it challenging for models to learn a consistent and generalizable representation of damage.

In summary, these findings confirm both the promise and limitations of optical-SAR modality for all-weather, global-scale disaster response. Although this combination performs well in events characterized by large-scale surface disruption (e.g., wildfires, volcanoes), it struggles with subtle or localized damage patterns. Incorporating richer data sources, such as fully

[Figure]

**Figure 9.** Typical failure cases of different models on Bata-Explosion-2021 and Noto-Earthquake-2024 in BRIGHT.

**Q4:** The manuscript currently lacks a comparison with optical-only baselines, which is crucial to highlight the value of multimodal methods. Readers may question whether the inclusion of SAR brings significant benefits and whether the additional cost of multimodality is justified. To avoid such doubts, I suggest adding experiments with optical-only inputs and comparing them with Optical+SAR results. This would further emphasize the unique value of the BRIGHT dataset and provide stronger evidence for the necessity of multimodal fusion.

**R4:** We sincerely thank the reviewer for this excellent suggestion. The question of how multimodal methods compare against an optical-only baseline is fundamental to justifying their value, and we appreciate the opportunity to provide a detailed clarification and new experimental evidence.

First, we would like to respectfully clarify the specific scenario that the BRIGHT dataset is designed to model. The core premise of BRIGHT is to facilitate **all-weather, rapid disaster response**. It is

constructed around the common and challenging real-world situation where a disaster event (e.g., a hurricane, flood, or wildfire) is followed by adverse atmospheric conditions (e.g., cloud cover, smoke) that prevent the timely acquisition of usable post-event optical imagery. Therefore, **the dataset's composition is intentionally pre-event optical + post-event SAR**. This represents a pragmatic and operationally vital workflow. Consequently, a direct **"optical-only" baseline** you suggested (i.e., pre-event optical + post-event optical) **is not feasible on the main BRIGHT dataset** by design, as high-quality post-event optical imagery is not a component of its primary structure.

However, we completely agree with the reviewer that a direct comparison is crucial for understanding the relative strengths of each modality when both happen to be available under ideal conditions. To address this valuable point, we have **conducted a new set of experiments on a specific subset of our data**, including Bata-Explosion-2020, Beirut-Explosion-2021, Hawaii-Wildfire-2023, Libya-Flood-2023 and Noto-Earthquake-2024, for which high-quality, cloud-free and preprocessed post-event optical imagery was also available to us.

We benchmarked three different setups on this subset:

➢ Optical-Only: Pre-event optical + Post-event optical

➢ SAR-Only (**BRIGHT's standard**): Pre-event optical + Post-event SAR

➢ Optical+SAR Fusion: Pre-event optical + Post-event optical + Post-event SAR

As the results show (**Table 8** in the revised manuscript), when high-quality, cloud-free post-event optical imagery is available, the post-event optical approach outperforms the post-event SAR approach across all tested models. For instance, using the state-of-the-art **DamageFormer** model, the **post-event optical setup achieves a final mIoU of 69.76%,** higher than the **65.56% from the post-event SAR setup**. This is expected, as optical imagery provides rich and intuitive visual information for damage assessment. Crucially, the results also demonstrate that fusing both modalities consistently provides the best results, outperforming even the strong post-event optical methods in every case. For example, DamageFormer's mIoU increases from 69.76% (post-event optical) to **70.79% with the addition of SAR data**, suggesting that SAR provides complementary information that can enhance the results even when high-quality optical data is present.

While this experiment provides a valuable benchmark, its results ultimately reinforce our motivation for BRIGHT. The high performance of the **optical-only model is entirely contingent on the availability of ideal, cloud-free post-event imagery**, a condition frequently not met in the critical window after many disasters. Therefore, **the small performance trade-off of the post-event SAR-based approach is justified by its invaluable all-weather, day-and-night operational capability**. BRIGHT is designed precisely to advance the development of models for these realistic, often non-ideal, but operationally critical scenarios. We have added this new experiment and discussion to the new Section in the revised manuscript to further emphasize the unique value of our dataset and the necessity of multimodal fusion.

Here, we show the revised part below for your convenience.

**Table 8.** Performance comparison of different post-event modalities on a subset of BRIGHT. Results are reported for UNet, DeepLabV3+, and DamageFormer on five disaster events where high-quality post-event optical imagery is available: Bata-Explosion-2020, Beirut-Explosion-2021, Hawaii-Wildfire-2023, Libya-Flood-2023, and Noto-Earthquake-2024.

| Method | Post-event modality | $F_1^{loc}$ (%) | $F_1^{clf}$ (%) | Final mIoU (%) | IoU per class (%) | | | |
|---|---|---|---|---|---|---|---|---|
| | | | | | Background | Intact | Damaged | Destroyed |
| UNet | SAR | 85.05 | 71.43 | 62.94 | 94.39 | 65.60 | 42.34 | 49.43 |
| | Optical | 86.46 | 75.64 | 65.96 | 94.56 | 68.62 | 45.27 | 55.36 |
| | Optical+SAR | 86.70 | 74.46 | 66.29 | 94.72 | 69.16 | 41.73 | 59.57 |
| DeepLabV3+ | SAR | 83.55 | 67.52 | 60.57 | 93.86 | 65.62 | 35.64 | 47.16 |
| | Optical | 85.79 | 74.39 | 64.87 | 94.33 | 67.90 | 44.31 | 52.94 |
| | Optical+SAR | 85.90 | 74.87 | 65.84 | 94.48 | 69.60 | 44.68 | 54.60 |
| DamageFormer | SAR | 88.41 | 73.43 | 65.56 | 95.30 | 70.62 | 41.31 | 55.00 |
| | Optical | 88.32 | 78.04 | 69.76 | 95.37 | 72.72 | 47.26 | 63.68 |
| | Optical+SAR | 88.86 | 79.27 | 70.79 | 95.56 | 73.64 | 48.44 | 65.51 |

**4.4 Impact of post-event modality on building damage assessment performance**

Although the primary design of BRIGHT is to facilitate all-weather disaster response through the use of pre-event optical and post-event SAR imagery, it is also important to understand how these modalities compare when high-quality post-event optical imagery is available. To this end, we conducted supplementary experiments on a subset of events, including Bata-Explosion-2020, Beirut-Explosion-2021, Hawaii-Wildfire-2023, Libya-Flood-2023, and Noto-Earthquake-2024, for which pre-processed post-event optical data were accessible. We evaluated three experimental setups: (i) optical-only (pre-event optical + post-event optical), (ii) SAR-only (pre-event optical + post-event SAR, i.e., the standard BRIGHT setting), and (iii) optical+SAR fusion (pre-event optical + post-event optical + post-event SAR).

Table 8 presents the experimental results. As expected, when ideal post-event optical imagery is available, the optical-only setup achieves higher performance than the SAR-only setup. For example, with DamageFormer, the optical-only configuration reaches a final mIoU of 69.76%, compared to 65.56% for SAR-only. Importantly, the performance gap between optical and SAR is not substantial, demonstrating that SAR alone provides a strong alternative in the absence of usable optical imagery. Moreover, the fusion of optical and SAR consistently yields the best results across all tested models. For instance, DamageFormer's mIoU further increases to 70.79% with Optical+SAR fusion, indicating that SAR contributes complementary information that strengthens performance even under optimal optical conditions.

These findings underscore two important insights. First, multimodal fusion is beneficial even when high-quality optical data are available, as SAR provides unique structural information that enriches the optical signal. Second, the performance of the SAR-only approach, being reasonably close to the optical-only results, highlights the practical value of SAR in real-world disaster scenarios where post-event optical imagery is often unavailable. BRIGHT is therefore designed to advance the development of models for these realistic, often non-ideal, but operationally critical all-weather disaster response settings.

**Q5:** The discussion of limitations and future directions is insufficient. At present, the conclusion mainly emphasizes the dataset's contributions, but it does not address its shortcomings in detail. It is suggested to include a separate subsection summarizing the limitations, such as the use of single-polarization SAR, the lack of time-series data, and the fact that most disaster events are concentrated after 2020.

**R5:** We sincerely appreciate this constructive suggestion to improve the discussion on the limitations of our dataset. We fully agree that a thorough and transparent account of the dataset's shortcomings

is essential for the community. Just for clarity, our original manuscript actually already included relevant content in Section 6 – Discussion:

➢ **Section 6.1 - Limitation of BRIGHT**: to address what we identified as the primary limitations, including potential registration errors, label quality, and sample/regional imbalance.

➢ **Section 6.2 - Significance of BRIGHT**: by suggesting that "Incorporating richer data sources, such as fully polarimetric SAR and LiDAR data, can further enhance the accuracy and reliability of future all-weather building damage assessments".

That said, we acknowledge that our initial discussion did not explicitly address two important points raised by the reviewer: the absence of time-series data and the temporal concentration of disaster events after 2020. We greatly appreciate this observation. In response, we have **revised Section 6.1 (Section 5.1 in the revised manuscript) to incorporate these limitations**, thereby providing a more comprehensive discussion. We believe these additions, prompted by the reviewer's insightful comment, have significantly strengthened the manuscript. We thank the reviewer again for helping us improve the quality of our work.

Here, we show the revised part below for your convenience.
* * *
**5   Discussion**

**5.1   Limitation of BRIGHT**

We begin this subsection by acknowledging that the composition of the BRIGHT dataset is fundamentally shaped by practical constraints in data availability. While BRIGHT represents a significant step forward in assembling a large-scale, multimodal, and globally distributed dataset for disaster response, it is important to recognize several inherent limitations. These limitations arise not only from the scarcity of open-access VHR SAR imagery, especially over disaster-affected regions, but also from the challenges of manual annotation and the uneven distribution of events. To provide a clearer picture for potential users, we summarize these constraints in four aspects below.

4. **Modality and temporal scope.** The dataset's scope is defined by two key characteristics of the available data. First, it exclusively utilizes single-polarization SAR imagery. The current version lacks the more informative multi-polarization or dense time-series SAR data, which, if available, could enable more nuanced damage characterization and long-term

31

recovery monitoring, respectively. Second, the dataset's temporal coverage is concentrated on events from 2020 onwards. This is a direct consequence of its reliance on modern commercial VHR SAR providers (Capella Space and Umbra), whose open-data initiatives largely commenced around that time.
* * *
**Q6:** The description of study areas and disaster events is somewhat redundant.

**R6:** Thank you so much for this valuable suggestion. In preparing the manuscript, we followed the style of other disaster-related papers published in ESSD, which typically provide detailed descriptions of

study areas and events. However, we agree with you that presenting too many event-specific details in the main text can be overwhelming and may distract readers from the core contributions of the work.

In response, we have **revised the structure of the manuscript to streamline this section**. Specifically, we have **moved the detailed descriptions of individual disaster events to the Appendix**, while retaining a concise overview in the main text. Specifically, the general description originally included in Section 2 has been merged into Section 3, now serving as its opening subsection. We believe this restructuring improves the readability of the manuscript by highlighting the key information while still making the detailed event descriptions accessible to interested readers.

Here, we show the revised part below for your convenience.
* * *
**2 Dataset Description**

**2.1 Study areas and disaster events**

145   We selected 14 disaster events across the globe for BRIGHT, as illustrated in Figure 2 and Table 2. Since both Capella Space and Umbra satellites were launched in 2020, we focused on study areas where disasters have occurred since then. The selected regions are primarily in developing countries, where public administration and disaster response capacities tend to be weaker compared to those in developed nations, making international assistance more critical. The dataset covers five major types of natural disasters: earthquakes, storms (including hurricanes and cyclones), wildfires, floods, and volcanic eruptions. Additionally, it includes human-made disasters, such as accidental explosions and armed conflicts. Detailed descriptions of the 14 disaster events are provided in Appendix A.
* * *
**Q7:** The terminology for "one-shot" may not be accurate. The authors describe it as using "a small number of labeled samples," which may be better defined as "few-shot." Since these concepts are borrowed from previous work, it is suggested to cite the corresponding references.

**R7:** We sincerely thank you for this precise and helpful comment. We agree that a clear and accurate definition of terms like "one-shot" and "few-shot" is crucial for methodological rigor.

The reviewer is correct that the phrase "a small number of labeled samples" generally corresponds to a "few-shot" learning scenario. Our intention in **using this more general phrase** in the original manuscript was to illustrate the practical context of disaster response, where it might be feasible for experts to quickly label a handful of examples from a new event. We recognize that this phrasing created an ambiguity between the real-world analogy and our specific experimental setup. Therefore, we **have modified the corresponding contents** in the revised manuscript.

Furthermore, we would like to take this opportunity to clarify a key distinction in our application of the "one-shot" paradigm. In many computer vision tasks, one-shot learning typically refers to learning to recognize new semantic classes from a single example. In our work, however, the classes (e.g., intact, damaged, destroyed) remain consistent across all disaster events. Our challenge is not learning new classes but rather adapting the model to a new data domain. That is a new, unseen disaster. Therefore,

we use the one-shot sample to facilitate cross-event adaptation, helping the model adjust to the unique visual characteristics, sensor properties, and damage signatures of the target event.

To ensure our manuscript accurately reflects this, we have revised the description to be more precise about both the terminology and the methodological context. Here, we show the revised part below for your convenience.
* * *
230      • One-shot setup: Recognizing the difficulty of the zero-shot setup, we introduce a one-shot setup. This setting simulates a realistic scenario where a single, representative sample from the new disaster can be quickly labeled to guide model adaptation. In this setting, a limited subset of labeled data (one pair for training and one pair for validation) from the target disaster event is incorporated into the training process. At the same time, the majority of the test set remains unseen. This setup evaluates the model's ability to leverage a minimal amount of manually labeled data to improve

235      disaster-specific adaptation.

It is worth noting that our cross-event transfer setup differs from classic few-shot learning tasks in the computer vision field (Amirreza Shaban and Boots, 2017; Wang et al., 2020). Our goal is not to recognize new classes, but to adapt the model's knowledge of existing classes to a new domain, *i.e.*, an unseen disaster event.
* * *
**Q8:** At line 420, the authors state that SAR is not sensitive to fine structural changes. Would this limitation reduce the value of multimodality in certain scenarios?

**R8:** Thank you for your thoughtful comment. We thank the reviewer for raising this insightful question. It is important to clarify that **every remote sensing modality captures only certain aspects of the Earth's surface, and each has its own strengths and limitations**. For example, optical imagery records reflect light in the visible and near-infrared spectrum, while SAR measures backscattered microwave signals. Consequently, each modality is inherently more or less sensitive to particular types of features.

In the case of single-polarization SAR, it is true that very fine structural changes/damages may not be well captured. However, **this limitation is not unique to SAR**. Indeed, even optical VHR satellite imagery can sometimes struggle to detect subtle or small-scale damage, as noted in [1]. Therefore, the challenge of capturing fine-grained structural changes/damages is a broader limitation of satellite data rather than a drawback that renders multimodality less valuable.

The core value of multimodal integration lies in ensuring operational continuity for disaster response, which is the primary motivation for our dataset. As discussed in the Introduction, optical EO data, while semantically rich, is frequently rendered unusable by cloud cover in the critical hours and days following a disaster. SAR data's all-weather capability is not just an advantage; it is often the only viable option for timely data acquisition. **The combination of pre-event optical data and post-event SAR data is therefore a pragmatic and powerful solution for rapid assessment,** even if neither modality alone is perfect.

Of course, in an ideal scenario, incorporating additional modalities such as fully polarimetric SAR, LiDAR, or multi-temporal imagery, would provide more complete coverage of structural damages. As

noted in Section 6.2 (Section 5.2 in the revised manuscript), we explicitly highlight this as an important direction for future dataset development.

In summary, while we acknowledge the limitations of single-polarization SAR for detecting fine-scale damage, this represents **a pragmatic trade-off rather than a fundamental flaw**. It does not diminish the value of multimodality but instead highlights the importance of combining complementary data sources to build robust, timely, and practical disaster response systems.

[6]  T. Manzini, P. Perali, J. Tripathi and R. R. Murphy, "Now you see it, Now you don't: Damage Label Agreement in Drone & Satellite Post-Disaster Imagery," *Proc. 2025 ACM Conf. Fairness, Accountability, and Transparency (FAccT '25)*, New York, NY, USA, pp. 1998–2008, 2025.

---

## Author Response (AR2)

Dear Topic Editor and Reviewers,

We sincerely thank you for acknowledging our work and for deciding our manuscript to enter [Publish subject to minor revisions (review by editor)] in Earth System Science Data (ESSD). We greatly appreciate your support, and the constructive guidance provided.

In response to the minor revisions, we have carefully revised the manuscript as requested. Please find below a summary of the updates made:

- a) As requested by the reviewer, we have now added a description of the SAR data pre-processing methodology, including a footnote to the source website, in the manuscript.
- b) Following the instructions, we have reviewed the accessibility of those mentioned figures for readers with color vision deficiencies. After checking using the Coblis Color Blindness Simulator we have revised Figure 11 to ensure its clarity for all readers.
- c) We have also added the necessary image sources and/or copyright information to the captions of the relevant figures.

We hope the revised version is now suitable for the next steps in the publication process. Should there be anything further required, please do not hesitate to let us know.

Kind regards,
Naoto Yokoya
on behalf of all co-authors